

# A semi-automatic procedure to support the detection of rapid-moving landslides using spaceborne SAR imagery

Giuseppe Esposito[1], Ivan Marchesini[2], Alessandro Cesare Mondini[2], Paola Reichenbach[2], Mauro Rossi[2], Simone Sterlacchini[3]

[1]National Research Council, Research Institute for Geo-Hydrological Protection (CNR-IRPI), Rende (CS), 87036, Italy
[2]National Research Council, Research Institute for Geo-Hydrological Protection (CNR-IRPI), Perugia, 06128, Italy
[3]National Research Council, Research Institute for the Dynamics of Environmental Processes (CNR-IDPA), Milano, 20126, Italy

*Correspondence to*: Giuseppe Esposito (giuseppe.esposito@irpi.cnr.it)

**Abstract.** The increasing availability of free-access satellite data represents a relevant opportunity for the analysis and assessment of natural hazards. The systematic acquisition of spaceborne imagery allows monitoring areas prone to geo-hydrological disasters, providing relevant information for risk evaluation and management. In case of major landslide events, for example, spaceborne radar data can provide an innovative solution for the detection of slope failures, even in case of persistent cloud cover. The information about extension and location of the landslide-affected areas may support decision-making processes during the emergency responses.

In this paper, we present a semi-automatic procedure, based on Sentinel-1 Synthetic Aperture Radar (SAR) images, aimed to facilitate the detection of rapid-moving landslides. The procedure evaluates changes of radar backscattered signals associated to land cover modifications, that may be also caused by mass movements. The procedure requires an initial manual selection of some parameters, and is able to execute automatically the download and pre-processing of images, the detection of SAR amplitude changes, and the identification of areas potentially affected by landslides, which are then displayed in a geo-referenced map. This map should help decision-makers and emergency managers to organize field investigations. The processes automatization is implemented with specific scripts running on a GNU/Linux operating system and exploiting modules of Open Source software.

We tested the processing chain, in back analysis, on an area of about 3000 km$^2$ in central Papua New Guinea that in February/March 2018 was struck by a severe seismic sequence that triggered numerous widespread landslides. In the area, we simulated a periodic survey of about seven months, from 12 November 2017 to 6 June 2018, downloading 36 Sentinel-1 images and performing 17 change detection analyses automatically. The procedure resulted in statistical and graphical evidences of widespread land cover changes occurred just after the most severe seismic events. Most of them can be interpreted as mass movements triggered by the main seismic shocks.


## 1 Introduction

Landslide recognition and mapping in rural areas represents one of the main challenges faced by the research community. The spatial and temporal distribution of landslides is well known mainly in urban areas, where they often cause severe
consequences to anthropic structures and population. On the contrary, landslides in rural and remote areas remain often unknown, limiting environmental evaluations like hazard and risk assessment (Guzzetti et al., 2012). Understanding where landslides have occurred may provide useful indications to forecast future events. In particular, the knowledge of the spatial distribution of landslides in a given region is essential to implement, calibrate and validate statistically and physically based methods (Rossi and Reichenbach, 2016; Mergili et al., 2014a; Mergili et al., 2014b) aimed to predict the possible location of
future mass movements or to identify areas where the probability of failure is negligible (Marchesini et al., 2014). As stated by Reichenbach et al. (2018), the quality and completeness of the landslide inventories may affect the reliability of the landslide susceptibility assessment (Steger et al., 2016). To produce inventory maps with limited errors and uncertainties (Santangelo et al., 2015), the mapping techniques should be selected taking into account a series of factors: the purpose of the inventory, the extent of the study area (Bornaetxea et al., 2018), the scale of the base maps, resolution and characteristics
of the available data, the skills and experience of the investigators, and the available resources (Guzzetti et al., 2000; van Westen et al., 2006, Casagli et al., 2017).

Besides conventional techniques (field mapping, visual interpretation of aerial photographs), remote sensing technologies based on satellite optical imagery, airborne/terrestrial laser scanning and digital photogrammetry represent innovative solutions for landslide detection and mapping. Recently, also multispectral and Synthetic Aperture Radar (SAR) satellite
images have been used with great success. To recognize landslides in multispectral and Very High Resolution (VHR) optical images, the most applied methods consist in the visual interpretation (Fiorucci et al., 2011; Ma et al., 2016) or in the semi-automatic classification (segmentation) that exploits different radiometric signatures of stable and failed areas (Martha et al., 2011; Mondini et al., 2011, Alvioli et al., 2018). However, optical images have some disadvantages and cannot be used when the analyzed areas are covered by persistent clouds or affected by shadow effects. To overcome these issues, SAR data
can represent an effective alternative, since they are not influenced by weather conditions.

Several techniques allow extracting information from SAR data to identify and map slope failures. The Differential Interferometric Synthetic Aperture Radar (DInSAR) (Gabriel et al., 1989) has been widely used to detect surface displacements over large areas with sub-centimeter accuracy. DInSAR is aimed to calculate phase differences between two or more multi-temporal images and has been successfully applied to analyze landslides (Calò et al., 2012; Zhao et al., 2012;
Cigna et al., 2013; Calvello et al., 2017; Tessari et al., 2017), earthquakes, subsidence, soil consolidation, volcanoes and tectonic deformations (Plank, 2014 and references therein). Other techniques exploit the amplitude information contained in the pixels of the SAR images. Amplitude of the backscattered signal is influenced by the type of target and varies according to several factors, such as the type of land use (e.g., water bodies, ice cover, forest type, bare soil), the surface roughness and the terrain slope. According to Colesanti and Wasowski (2006), amplitude SAR imagery represents potentially a very useful



source of information, which can complement high resolution optical imagery and aerial photography in feature detection. Generally, amplitude-based methods analyze the correlation of the speckle pattern of two images (e.g., pre- and post-event, where the terms "event" can refer to a major natural and/or human-induced hazard affecting a given area, as an earthquake, an hurricane, a forest fire, etc.) to map the land cover changes (Raspini et al., 2017). To date, the landslide mapping community has shown a poor attitude in using this type of product, so that only few studies have demonstrated the valuable

contribution of SAR amplitude changes in landslide detection and mapping. According to Mondini (2017), this is due to a series of problems and drawbacks represented by: 1) the complex pre-processing procedures; 2) the acquisition geometry that can affect the quality of the images over mountainous areas where landslides are likely to occur; 3) the difficulty in using the SAR signal in traditional statistical classification approaches mainly due to speckling. A successful example of the use of amplitude variations of the radar signal to analyze landslides is described by Zhao et al. (2013), which inferred the

Jiweishan rock slide in China using changes in SAR backscattering intensity in ALOS/PALSAR images. Tessari et al. (2017) verified that when the phase information cannot be exploited, amplitude of the reflected signal is very useful to detect and map rapid-moving landslides that cause significant variations in the ground morphology and land cover. Mondini (2017) proved that both landslides and flooded areas can be detected by verifying changes in the spatial autocorrelation in a multi-temporal series of SAR images. Konishi and Suga (2018) also identified a series of landslides in Japan by analyzing intensity

correlation between pre- and post-event SAR images.

Besides the described techniques, recent advances in SAR technology are promoting the use of polarimetric SAR data (PolSAR) characterized by full-polarimetric information (i.e., acquired in single polarization, dual polarization, and fully polarimetric modes) for a target in the form of the scattering matrix (Skriver, 2012). According to Plank et al. (2016), these data provide more information on the ground, which enables a better land cover classification and landslide mapping.

Successful applications were described by Yamaguchi (2012), Shimada et al. (2014), Li et al. (2014) and Plank et al. (2016). The use of SAR data to analyze landslides and/or potentially unstable slopes should hence increase, also in relation to a series of valuable technical innovations. The improved revisiting times and spatial resolution of the images, for example, represent a key factor during disaster response operations, when a preliminary localization of areas potentially affected by major landslides is crucial. Revisiting times have been in fact reduced from 35 days of ERS and Envisat satellites, to 12

hours (at 40°latitude, in case of emergency response) of the COSMO-SkyMed constellation (Casagli et al., 2017). The enhanced spatial resolution (azimuth x range) of images spans in the order of few meters (i.e. 1-10 m), resulting more detailed with respect to the resolution of the first-generation satellites characterized by pixel sizes of 10-30 meters (Plank, 2014).

Among the most advanced SAR spaceborne systems (Casagli et al., 2017), there are those of the mission Sentinel-1 operated

by the European Space Agency (ESA) in the frame of the European Union's Copernicus Programme. Satellites Sentinel-1A and 1B acquire images characterized by a spatial resolution up to 5x5 m, depending on the acquisition mode, and a temporal resolution ranging from 6 to 12 days according to the surveyed geographic area. The Sentinel-related products have a global coverage and are freely available to all users registered on the ESA data hub (https://scihub.copernicus.eu/). This is a



considerable benefit that is leading many research institutions and public administrations to use Sentinel data to investigate

landslides and other natural processes (Salvi et al., 2012; Dai et al., 2016; Twele et al., 2016; Intrieri et al., 2018). According to Raspini et al. (2017), the future increased number of available satellites characterized by shorter revisiting times and high spatial resolution will offer relevant information for decision support and early warning systems. Currently, significant limitations concern the real-time and/or quasi real-time detection of rapid flow-like mass movements, rock failures, and flash floods characterized by evolution times ranging from minutes to hours. This poses a challenge for the geo-hydrological risk

management based on satellite technologies.

In this article, we present a semi-automatic procedure aimed to support the detection of rapid-moving landslides by performing the periodic survey of unstable slopes, using spaceborne radar imagery. We focus on rapid-moving landslides since they determine evident land cover changes with respect to slow-moving failures. The main purpose of the implemented procedure is to emphasize areas where evident land cover changes (potentially related to slope failures) occur, facilitating the

following possible phases of mapping and/or field survey. In other words, the procedure allows producing a map where pixels are ranked based on the level of change observed by comparing two consecutive satellite images. Decision makers and emergency managers can use this map to organize possible verifications and field investigations.

The procedure is implemented in a processing chain based on free data and software, and exploits radar backscattered signals recorded within the Sentinel-1 SAR images. The values of some parameters related to the used algorithms must be provided

by the user. In alternative, they can be set based on the values derived from other similar areas. The processing chain was applied, in back analysis, to an area in Papua New Guinea that in February/March 2018 was struck by a severe seismic sequence, which triggered numerous widespread landslides.

## 2 Methodology

### 2.1 Pre-processing of SAR images

The implemented procedure is based on Sentinel-1 images available in Level-1 Single Look Complex (SLC) mode, with a VV-VH polarization and Interferometric Wide acquisition mode. Level-1 SLC products are images provided in slant range geometry, georeferenced using orbit and attitude data from the satellite. Each image pixel is represented by a complex magnitude value and contains both amplitude and phase information (ESA, 2018). Pre-processing of the images is performed using the Graph Processing Tool (GPT) of the Sentinel-1 Toolbox[1], and includes the following steps: (1) thermal noise

removal, (2) radiometric calibration, (3) Topsar de-burst, and (4) multi-looking processes.

The thermal denoising consists in the removal of dark strips with invalid data from the original data. This operation is performed with the SNAP algorithms by subtracting the noise vectors provided by the product annotations from the power detected image (ESA, 2017); the radiometric calibration allows computing the slant-range radar brightness coefficient ($\beta_0$)

---

[1] GPT is the Command Line Interface of the Open Source software SNAP Sentinel Application Platform, version 6.0 - http://step.esa.int/main/toolboxes/snap/. The source code of SNAP is available at https://github.com/senbox-org





(El-Darymli et al., 2014) by converting digital pixel values in a radiometric calibrated backscatter; the Topsar de-burst

removes black-fill demarcations between the single bursts forming sub-swaths of the IW-SLC products, allowing retrieving

single images; and multi-looking is carried out to reduce the standard deviation of the noise level and to obtain

approximately square pixels of about 14 m (mean ground resolution), by applying a factor of 1:4 (azimuth:range).

Consecutive SAR images, selected to detect amplitude changes of the radar signal (i.e. change detection), are co-registered

with a DEM-assisted procedure that uses the Shuttle Radar Topography Mission (SRTM) 1 Sec digital elevation model

(DEM), auto-downloaded by SNAP. After the co-registration, the resulting stacked images are filtered for speckling

reduction using the adaptive Frost filter (Frost et al., 1982), with a filter size in X and Y of 5 pixels, and a damping factor

(defining the extent of smoothing) of 2.

## 2.2 Detection of SAR amplitude changes

To perform the change detection analysis, the Log-Ratio (LR) index is calculated as described by Mondini (2017). This

index measures the change in the backscattering that might be induced by land cover changes related to both natural (e.g.,

landslides, floods, snow melting) or human-induced processes (e.g., mining activities, deforestation), in a defined time

interval. For each pair of corresponding pixels belonging to consecutive pre-processed SAR images, the Log-Ratio index is

calculated as follows:

$$LR = \ln \left( \frac{\beta_{0,i}}{\beta_{0,i-1}} \right) \qquad (1)$$

where $\beta_0$ is the radiometric calibrated backscatter (i.e., SAR amplitude), and i, i-1 indicate two consecutive pre-processed

SAR images. For each pair of pre-processed images, a LR layer is computed, and related pixels can be characterized by

positive or negative values, depending on the backscattering changes.

When the study area (i.e. Area of Interest - AoI) corresponds to a zone smaller than the entire LR layer, a subset is extracted

by using the subset tool in SNAP.

## 2.3 Segmentation of the Log-Ratio layer

The segmentation of the LR layer is aimed to group pixels with similar LR values into unique segments. The process is

performed with the i.segment module in GRASS GIS 7.4 (Momsen and Metz, 2017), using the "Mean Shift" algorithm and

the adaptive bandwidth option.

The first step of the Mean Shift algorithm consists in the smoothing process of the LR layer. To do this, the algorithm

requires from the user the definition of the following parameters: (i) the initial bandwidth size (*hr*); (ii) the spatial kernel size

(*hs*); (iii) the threshold (*th*), and (iv) the maximum number of iterations. We acknowledge that the smoothing considers the

pixel *p* (having value $LR_p$) in the center of a spatial kernel of size *hs* and assigns to this a mean value calculated using only




the pixels that are inside the spatial kernel, with values ranging between (LR$_p$–hr) and (LR$_p$+hr). The unit of measurement of

$hs$ is in pixel and $hr$ is a range of LR values. In other words, the smoothing allows that each pixel value is computed considering all pixels not farther than the spatial kernel ($hs$) with a difference not larger than $hr$. This means that pixels that are too different from the considered pixel $p$ are not included in the calculation of the new value.

With the adaptive option, for each pixel $p$, $hs$ is fixed whereas the bandwidth size $(hr_{ad})_p$ is recalculated to account for the variation of the pixels values (LR in this work) across the spatial kernel centered in $p$. The aim is to avoid the drawbacks of a

global bandwidth consisting in under- or over-segmentation. More in general, the adaptive bandwidth size ($hr_{ad}$) is calculated using the following equation:

$$(hr_{\mathrm{ad}}) = avgdiff \cdot \exp\left(-\frac{avgdiff^2}{2 \cdot hr^2}\right) \qquad (2)$$

where $avgdiff$ is the average of the differences between the value of the central pixel and the values of other pixels included in the kernel; $hr_{ad}$ is maximum if the $avgdiff$ is equal to the user-defined $hr$, which is also the upper limit of the possible $hr_{ad}$ values (i.e. $hr_{ad}$ is always smaller than $hr$). The adaptive option is particularly useful when data are characterized by high and abrupt spatial variability (as is the case of the LR layers), and a smoothing preserving the main discontinuities is required (Comaniciu and Meer, 2002).

The Mean Shift algorithm recalculates the central pixel values until a user-defined maximum number of iterations is reached, or until the largest shift (value difference) resulting between the central pixel and the pixels inside the kernel is smaller than a threshold ($th$) defined by the user. The threshold must be bigger than 0.0 and smaller than 1.0: a threshold of 0 would allow only pixels with identical values to be considered similar and clustered together in a segment, while a threshold of 1 would allow everything to be merged in a very large segment (Momsen and Metz, 2017). A more or less conservative threshold

needs to be chosen considering the spectral properties of the analyzed image. After the smoothing, pixels in the range of the estimated local maxima (Comaniciu and Meer, 2002), which are close to each other, are clustered and included in a new map containing the defined segments. To reduce the "salt and pepper effect", the segments containing less than a preferred minimum number of pixels are eliminated, by specifying the *minsize* parameter within the i.segment command.

To select the appropriate parameter values (i.e. tuning), a specific analysis should be carried out interactively (manually)

before the implemented procedure is started. In particular, variability of the segmentation outcomes to the usage of different values for the $hs$, $hr$ and $th$ parameters must be analyzed. This analysis is event-dependent because it can be executed using consecutive SAR images acquired before and after a well-known landslides event occurred in the past, in the area to be surveyed or in areas which are considered similar by geomorphologists, based on the types of land cover and expected types of landslide. The spatial kernel size $hs$ can be heuristically chosen according to the size of the land cover changes that should

be detected. Keeping constant the spatial kernel size, $hr$ and $th$ values can be changed iteratively, evaluating the results in




terms of number and sizes of segments generated by the Mean Shift algorithm. As general rule, one can expect that large values of *hr* will correspond to few (but big) segments, whereas small values of *hr* will determine many small segments. This is due to the fact that smoothing increases when larger values of *hr* are used. The effect of the variation of the value of *th* is expected to work in the opposite direction but being much less effective on the segmentation outcomes. The first scenario

(few and very large segments) is useless since it cannot be used for geo-localize the possible land cover changes. The second scenario (many and small segments) is the result of the segmentation of the random noise of the back-scattered SAR images and it is, again, useless. We assume that a possible criterion for selecting the best values of *th* and *hr* is to search for the combination of values that optimize, at the same time, the number of segments and their average size with respect to the expected land cover changes. An example of the procedure for the selection of the best values for the *th* and *hr* parameters is

described in section 3.

**2.4 Identification of areas potentially affected by land cover changes**

After the segmentation step, a statistical analysis of the LR values included in each segment is carried out to identify segments that, with high probability, are related to significant land cover changes.

For each segment, the arithmetic mean ($\mu_s$) of the included LR pixel values is calculated as follow:


$$\mu_s = \frac{1}{k}\left(\sum_{j=1}^{k} p_j\right) \tag{3}$$

where *s* indicates the segment, *p* the pixel value and *k* the number of pixels in the segment. We define as "average layer", the raster layer where at each segment is associated the corresponding value of $\mu_s$. Afterwards, in order to filter segments and

extracting only those representing significant statistical changes, the $\mu_s$ values have to be compared with reference $\mu$ and average standard deviation ($\bar{\sigma}$) related to no-change conditions. These reference figures have to be calculated before the initialization of the processing chain and after the segmentation of the event-related LR image, using preceding SAR images when no heavy rainfall, landslides and earthquakes occurred, by applying the following formulas:

$$\mu = \frac{1}{m}\left(\sum_{j=1}^{m} p_j\right) \tag{4}$$






$$\sigma_s = \sqrt{\left(\frac{1}{k-1}\right)\left(\sum_{j=1}^{k} p_j - \mu_s\right)^2} \tag{5}$$

$$\bar{\sigma} = \frac{1}{r}\left(\sum_{z=1}^{r} \sigma_{s_z}\right) \tag{6}$$

where $m$ is the total number of pixels in the pre-event LR image used as reference, $p$ is the related pixel value, and $r$ is the total number of segments derived from the segmentation of the event-related LR image, characterized each one by a $\sigma_s$.

In this way, both the $\mu$ and $\bar{\sigma}$ are calculated in a kind of "warm-up stage" of the described processing chain. Generally, suitable reference figures can be calculated since the generation of three or four reference LR layers characterized by values normally distributed (i.e. Gaussian), indicating a random nature of the LR distribution that is typical of no significant land cover changes. We highlight that, in such a case, given that LR values are typically small and positive or negative, the $\mu$ value is equal or very close to zero.

In this way, all the segments of the "average layer" characterized by $\mu_s$ values larger than $|\mu+(2\bar{\sigma})|$ are then extracted and classified. Segments with $\mu_s$ values lower than a confidence interval of 95% ($\mu_s < |\mu+(2\bar{\sigma})|$) are instead discarded. Segments where $\mu_s$ is greater than $|\mu+(2\bar{\sigma})|$ and smaller than $|\mu+(3\bar{\sigma})|$ are reclassified to the integer value of 2. Similarly, the values 3 and 4 are used to classify segments with $\mu_s$ values included in the range $|\mu+(3\bar{\sigma})|$ to $|\mu+(4\bar{\sigma})|$, and larger than $|\mu+(4\bar{\sigma})|$, respectively. All these segments form a new raster layer representing a map of areas characterized by relevant SAR

amplitude changes including those affected by rapid slope movements.

In order to refine this map, all the segments with the same values (i.e., 2, 3, or 4), that are spatially contiguous and are formed by at least a user-defined minimum arbitrary size in terms of pixels (i.e. minimum detectable landslide area) are merged together, and the following statistics are then computed: 1) count of merged segments; 2) maximum number of pixels included within a single segment; and 3) average number of pixels included within a single segment.

The final segment map produced by the semi-automatic chain is georeferenced in the WGS84 reference system (EPSG 4326) by means of the Terrain correction tool of SNAP.

## 2.5 Script implementation

Besides segmentation of the event-related LR image with tuning of related parameters, and the calculation of reference μ and $\bar{\sigma}$ related to no-change conditions, all the other described procedures have been implemented in two groups of scripts that

can be executed automatically.

The python-based script (Fig. 1, Data ingestion) is devoted to the automatic querying and downloading of Sentinel-1 SAR images from the ESA Sentinel Data Hub. The script, based on the SentinelSat toolbox (Kersten et al., 2018), is set to query



the Sentinel Data Hub with a daily frequency even though new images may be available every 6 or 12 days, depending on the geographic area.

The group of scripts, written in GNU/Bash programming language (Fig. 1), is aimed at: (i) pre-processing the Sentinel-1 images (section 2.1), (ii) detection of the changes in SAR amplitude and production of Log-Ratio maps (section 2.2), (iii) segmentation of the LR maps (section 2.3) and, (iv) identification of areas potentially affected by land cover changes (section 2.4). This group of scripts is executed automatically when new Sentinel-1 images are available and downloaded by the python-based script.

The bash-scripts require the following settings defined by the user: 1) the path of the folder where the downloaded SAR images are stored; 2) values of the parameters required for the segmentation (see section 2.3), and (3) the spatial coordinates of the area of interest (if it is a portion of the downloaded SAR images). No further information is needed since the commands are executed in a unique automatic sequence. To survey the same area for an unlimited time period, all these settings have to be defined only for the chain initialization.

**3 The Papua New Guinea test site**

We selected as test site, an area located in central Papua New Guinea (Fig. 2) that was affected by a severe seismic sequence at the beginning of 2018. On 25 February, the area was hit by a main seismic event (M7.5) followed by several aftershocks, including a M6.7 earthquake on 6 March. The strong mainshock, rather superficial with a hypocentral depth at 23.4 km (USGS, 2018), caused building collapses, road damage and widespread landslides mostly along the Tagari river valley and

the slopes of Mount Sisa. According to the International Federation of Red Cross and Red Crescent Societies (IFRC, 2018), more than 100 people died, most of them due to landslides.

To test the implemented procedure, we have analyzed an area of about 3000 km$^2$ in the mountainous region close to the epicenters of the mainshock (AoI in Fig. 2), where preliminary information on landslides were available (Petley, 2018a,b).

To simulate a periodic survey covering pre- and post-earthquake periods, we downloaded 36 Sentinel-1 images from the

Sentinel Data Hub (https://scihub.copernicus.eu/) acquired along the satellite track n.82 in ascending orbit with a temporal frequency of 12 days, from 12 November 2017 to 6 June 2018. The downloaded images were used to perform a total of 17 change detection analyses which resulted in likewise LR layers, with a pixel size of about 14 m. The values of the segmentation parameters were defined with an interactive manual analysis (see section 2.3) by segmenting the "pre-post M7.5 earthquake" LR layer, selecting the spatial kernel size (*hs*) of 10 pixels (see section 2.3), and setting the maximum

number of iterations to 200. This size of the spatial kernel was set to 10 pixels to detect significant differences of LR values during the smoothing stage of the segmentation process, taking into account the approximate expected size of the land cover changes. In the interactive (manual) analysis, we selected bandwidth sizes (*hr*, see section 2.3) ranging from 0.0005 to 0.016, and thresholds (*th*) from 0.001 to 0.016 (Fig. 3), obtaining 20 different parameter combinations. For each couple of parameters, the number of generated segments and their average size were plotted as shown in Fig. 3. Points highlight the



major impact of the *hr* parameter with respect to the role played by the threshold (*th*) parameter, in defining the number of total generated segments. Below an *hr* value of 0.004 over segmentation occur, whereas for *hr* values equal or larger than 0.004, the number of generated segments tends to become small and constant. With the aim of avoiding over segmentation while maintaining a reasonable average size of the segments (to be able to delineate also small patches of the terrain where changes occurred), and considering a visual inspection of the segmentation results, we decided to run i.segment in the semi-

automatic processing chain using the following set of parameters values: *hs*=10, *hr*=0.004, *th*=0.008, *minsize*=2, *iterations*=200 (see section 2.3).

After the segmentation of the 17 LR layers, the segments with a minimum size of 5 pixels were extracted in the area of interest (an example in Fig. 4d), and statistics were calculated according to the confidence intervals described in the methodology section. We decided to select only the segments with a minimum size of 5 pixels, corresponding to a minimum

area of 980 m$^2$ (i.e. 196 m$^2$ · 5), after a rough evaluation of the preliminary landslide-related images published on news websites and social networks, and considering that the occurrence of smaller mass movements in the test area were not significant at the scale of our analysis.

In Fig. 5, statistics of the selected segments are displayed for each change detection. The analysis of the histograms revealed that two main peaks occurred for the change detections 9 and 10. Change detection 9 considers images acquired before and

after the M7.5 earthquake, whereas change detection 10 the images acquired on 28 February and 12 March 2018. The first peak highlights widespread changes related also to landslides extensively documented after the M7.5 event (Petley, 2018a). The second peak was instead unexpected and was probably due to the occurrence of further landslides triggered by the M6.7 event on 6 March 2018. In Fig. 6, segments related to these two peaks are displayed (red pixels = change detection 9; blue pixels = change detection 10). The map shows that the two groups of segments are in general accordance with the areas (in

yellow) really affected by landslides, as interpreted from the optical images available on the Planet explorer application (Planet, 2017).  Small widespread segments outside the landslide areas, mostly related to stream changes and noise, also resulted in the AoI. Similar random segments occurred in the pre- and post-event change detections, as also displayed by statistics in Fig. 5.

**3.1 Statistical evidence of landslide-like segments**

Clear evidence of the widespread land cover changes induced by the two earthquakes, as well as the timing of their occurrence, resulted also from a statistical analysis of the segment areas derived from each change detection. The comparison of some representative statistics of segment's areas related to different change detections are shown in Fig. 7. For each comparison, four types of statistics are displayed: Quantile-Quantile (Q-Q) plot, Empirical Cumulative Distribution Function (ECDF), Density plot, Frequency plot.

The first column in Fig. 7 compares areas of the segments resulted from the change detections 11 and 7, which we assume not being affected by landslides triggered by the earthquake (i.e., NO-EVENT/NO-EVENT, where the term EVENT refers to the earthquake shocks); the second column shows the comparison between areas of the segments resulted from change





detections 6 and 10, with the second including seismic-triggered landslides (i.e., NO-EVENT/EVENT); the third column indicates the comparison between areas of the segments from the change detections 9 and 10, both with landslides (i.e.,

EVENT/EVENT).

Clear differences can be noted between the areas distribution of segments with and without landslides (change detection 6 - change detection 10) displayed in the NO-EVENT/EVENT plots. This difference is particularly highlighted in the Q-Q plot, where the gap between the blue line representing the real distributions and the theoretical similarity condition (red line) is evident.

Taking into account the p-values from the Kolmogorov-Smirnov tests carried out for all the 136 distribution comparisons (the combination of 17 change detections taken 2 at a time without repetition), it arises a general similarity (i.e. p-value > 0.05) among 86 of the 95 distributions that were not affected by landslides (NO-EVENT/NO-EVENT) (Fig. 8). On the other hand, among the 30 NO-EVENT/EVENT distributions, the majority (21/30) are different. The EVENT/EVENT distribution is also different (Fig. 8).

We have attempted to analyze the distribution of the areas of the segments derived from the change detections to verify if they follow the empirical statistical distribution of the landslides size, as described by different theoretical models (Stark and Hovius, 2001; Malamud et al., 2004; Rossi et al., 2012; Schlögel et al., 2015).

By using the tool implemented by Rossi et al. (2012) for estimating the probability distribution of landslide areas, we verified that medium and large-scale areas of segments obtained with the change detections 9 and 10 followed a landslides-

like behavior. In fact, the two probability distributions obtained with a Double Pareto Simplified model (Fig. 9) resulted in inverse power law decays for medium and large areas, highlighting a moderate agreement with empirical data. A rollover (inflection) in correspondence of small areas (e.g., Malamud et al., 2004) however is not present. This can be due to a consistent detection of small changes (i.e. ~ 1000 $m^2$) ascribable to landslides and other random land cover modifications, other than some noise. In addition, it is worth remembering that segments with areas $\leq 980$ $m^2$ were not considered.

As shown in Fig. 9, the two analyses did not pass the Kolmogorov-Smirnov test (p-value = 0). This may be explained with the fact that the input datasets (i.e. segments area) were not obtained with a proper landslide mapping activity, as described in Guzzetti et al. (2012), but with a semi-automatic procedure that is not aimed at landslide mapping operations, as clarified in the Introduction section, but at identifying land cover changes also related to landslides. Although this, results seem to be fair consistent with curves of proper landslide inventory data (Schlögel et al., 2015).

**4 Discussion**

In this article, we describe a semi-automatic processing chain aimed at identifying SAR amplitude changes that can be partially explained by the occurrence of rapid mass movements. We have selected SAR data since they have the advantage to be not affected by the cloud cover disturbance. In fact, as described by Mondini et al. (2019), the use of SAR amplitude data can mitigate the cloud coverage issue and can allow detecting landslides that, otherwise, might remain unknown or





unnoticed for a long time. In this way, the procedure can be exploited for a "continuous", in terms of time, slope monitoring activity, even if failures occur during long-lasting periods of precipitation and persistent cloud cover that do not allow to use optical data for a rapid and detailed landslide recognition. In the selected study area, a widespread cloud cover persisted for several weeks during and after the seismic sequence. The first cloudless optical image of the area damaged by the seismic shaking was published by the daily monitoring service delivered by ©2019 Planet Labs Inc. (www.planet.com) on 25 March,

almost one month after the M7.5 mainshock that triggered numerous landslides. The high cloud persistence is quite common in Papua New Guinea, and in fact this is included in the cloudiest regions of the world with annual cloud frequency (proportion of days with a positive cloud flag) higher than eighty percent (Wilson and Jets, 2016; Mondini et al., 2019). As consequence, the use of optical data in this area, and in other mountainous regions exposed to prolonged rainfall related to monsoons, cyclones or other persistent meteorological systems results tricky.

The tuning of the segmentation parameters is the key element for identifying areas affected by significant land cover changes, also induced by rapid-moving slope movements. This process can be retained event-dependent, requiring a well-known landslide event occurred in the past in the analyzed area or in zones with similar topographic and land use characteristics. In the case study here described, definite values of the segmentation parameters were obtained by segmenting the pre-post M7.5 earthquake LR layer, by testing different values combinations (Fig. 3). This may represent a limit of the

proposed procedure if one would apply it in different geomorphic settings without past landslide events, or identifying different types of slope failures. On the other hand, if a proper event-based tuning operation is performed, a continuous monitoring of slopes can be efficiently carry out without temporal limitations, exploiting both pre- and post-event available images, as done in the current case history. The described application highlighted in fact that by keeping the same parameters values, landslides and other land cover changes triggered by the M6.7 aftershock were also detected. The

occurrence and location of these secondary failures (blue pixels in Fig. 6) were not known before our analysis because not reported by news and local government websites, and also missing in the maps of the Copernicus Emergency Management Service (https://emergency.copernicus.eu/mapping/list-of-components/EMSR270) activated for the disaster response. The general lack of information related to these failures was likely due to a series of issues affecting both the field and the satellite surveys in the aftermath of the M6.7 earthquake. In fact, an effective assessment in the field was impeded by the

road damages caused by the mass movements triggered by the previous major M7.5 event, whereas the use of optical satellite images was hampered by a widespread cloud cover that, as stated before, persisted during several weeks after the two main seismic shaking. The first information about the occurrence of these landslides were provided online by Petley (2018b), about one month later, without a clear indication of their relationships with the M6.7 earthquake. The detection of this second set of failures in areas poorly affected by slope movements triggered by the M7.5 event demonstrates the relevant

usefulness of the proposed processing chain.

The segments located outside the landslide-affected areas are caused by other land cover changes that are out of the aims of this study, or by random noise effects. Segments related to these changes can be easily identified because composed by an



average number of pixels close to ten, as detected in all the change detections, whereas segments related to landslides (i.e. change detections 9 and 10) are characterized by a higher number pixels (Fig. 5c).

A suitable segmentation can allow hence to get statistical evidences of event landslides occurrence. Statistical distributions of the three parameters shown in Fig. 5 provide distinctive signatures of widespread land cover changes triggered by the M7.5 mainshock and by the M6.7 aftershock. It is worth noting, however, that 95-percentiles highlighted in the plots are exceeded also by other peaks (e.g., change detection 13 in the segment count), that cannot be considered as diagnostic of landslide occurrence since they are ephemeral, and are not steady in all the three plots as the change detections 9 and 10. In

case of small-scale landslides occurring in localized portions of a wide area, the related statistical signals may result imperceptible if these are of the same magnitude of other previous and successive signals not related to landslides. In cases like this, distinctive evidence of slope failures can be achieved by starting the process chain with a smaller subset of the LR layer (i.e. monitored area).

Overlapping between the segments (i.e. change detections 9 and 10) to ground truth data revealed that largest SAR amplitude

changes correspond often with landslides (Fig. 6). A further evidence was provided by the statistical distributions shown in Fig. 9, resulted similar to those estimated by other landslide-related studies (Stark and Hovius, 2001; Malamud et al., 2004; Rossi et al., 2012; Schlögel et al., 2015).

The outcomes of this study represent a concrete example on how to exploit the relevant advantages of Open Source software with a command line interface (i.e. SNAP and GRASS GIS) to implement automatic processing chains. Moreover, it is

worth noting that the proposed methodology can be properly adopted to monitor areas in the order of thousands of square kilometers if powerful hardware resources are available. In fact, the pre-processing and segmentation steps require significant amounts of calculation power and memory. It is well known that the Mean Shift is a time-consuming algorithm for large datasets (Wu and Yang, 2007), and convergence for large areas can be reached in dozens of hours. Segmentation times are proportional to the dimensions of the monitored area, and to the selected spatial kernel size ($hs$).

A final remark concerns the occurrence of landslides in the study area. Generally, landslides in the mountainous sectors of Papua New Guinea are very common processes. Earthquakes with a magnitude greater than 5 are among the dominant factors triggering widespread landslides. According to Robbins and Petterson (2015), such earthquakes occur regularly in the country but records of the triggered landslides are surprisingly lacking. The lack of systematic reporting and the remoteness of communities affected by such events, also impeded an adequate characterization of landslide hazard and risk (Blong,

1986). Robbins et al. (2013) stated that landslides occur annually, and failures tend to range from few cubic meters of material to mass movements with estimated volumes of $1.8 \times 10^9$ m$^3$, varying from debris slides, avalanches and flows to translational and rotational slides. In this framework, the landslide detection procedure described in the article may result a relevant tool for local authorities of countries characterized by extensive remote areas repeatedly affected by slope failures, and for the humanitarian organizations operating in response to geo-hydrological disasters.



## 5 Conclusions

This study presented a semi-automatic procedure aimed to support the detection of rapid-moving landslides on vast mountainous areas. It is based on SAR data acquired systematically by the Sentinel-1 satellites. The computation of the Log-Ratio index and segmentation of the consequent raster layers allow detecting areas affected by multi-temporal variations of the radar backscattered signal. Among them, areas potentially related to rapid-moving landslides are identified with a robust statistical analysis. The performance of the implemented procedure was tested in back analysis on an area of about 3000 km$^2$ in Papua New Guinea. Here, in 2018, two consecutive earthquakes (M7.5 and M6.7) triggered widespread slope failures causing more than 100 fatalities and severe damage to roads and buildings. The simulation of a multi-temporal survey of about seven months, before and after earthquakes, revealed the ability of the implemented procedure to detect statistical evidences of significant land cover changes in correspondence of the two events. Moreover, results demonstrated that the zones affected by landslides were identified with a fair accuracy, as compared to the ground truth data.

The study highlighted advantages of free SAR products that may guide the scientific community and the local authorities to develop archives of freely accessible data, suitable for implementing streamlines of information aimed to monitor natural and urbanized areas. As demonstrated in the case study, the proposed procedure has the potential to be a valid support in landslide emergency management, providing in near real-time relevant information for civil protection authorities and scientists involved in the emergency response. Future improvements may limit the user decisions in the model parameterization, optimizing the processing times and refining the filtering of landslide-related changes by considering also geological and geomorphological factors.

*Author contributions*. IM, GE and AM designed and realized the processing chain. GE and IM carried out the experiments. GE wrote a first draft of the manuscript. MR and GE carried out statistics about landslide-like segments. IM, GE and PR analyzed the results. GE, SS, IM and PR improved the final manuscript version.

*Competing interests*. The authors declare that they have no conflict of interest.

*Acknowledgements*. The research was developed in the framework of the STRESS (Strategies, Tools and new data for REsilient Smart Societies) project, funded by Cariplo, focusing on designing, implementing and testing of a Spatial Information Infrastructure, enabling the provision to spatial planners and risk managers of new data and tools to improve susceptibility, hazard and impact assessment of geo-hydrological phenomena.
(http://www.idpa.cnr.it/progetti_STRESS_ufficiale.html).

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





**Figure 1: Flowchart of the automatic steps of the processing chain described in the text. The single steps are grouped in two main scripts (developed using Python and Bash scripting languages). I(t₁) and I(t₂) represent two consecutive SAR images (or set of images).**





**Figure 2: Location of the test site. The Area of Interest (AoI) is shown with a black rectangle in the main map and with a red triangle in the inset.**


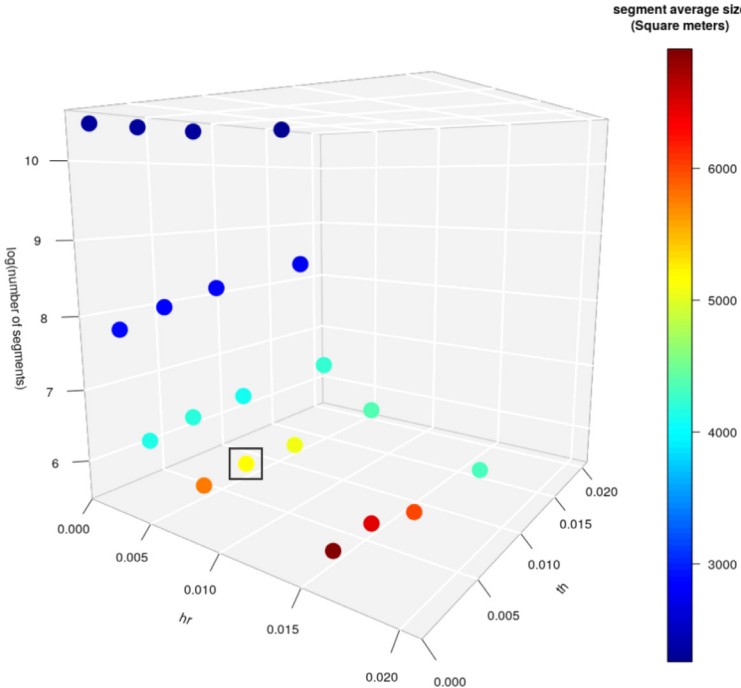

**Figure 3 - Data analysis aimed at evaluating the best combination of bandwidth (*hr*) and threshold (*th*) values. The black rectangle identifies the selected combination.**

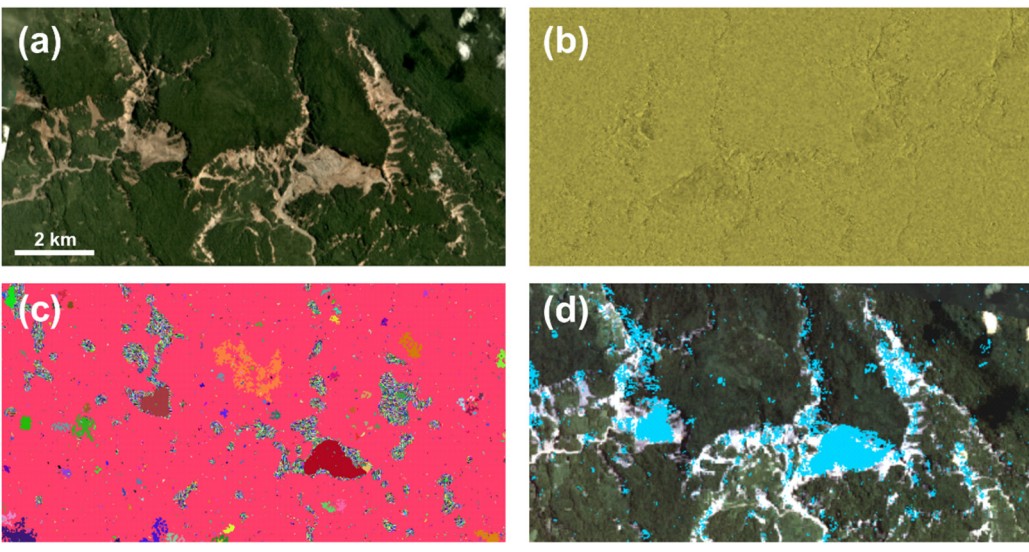

**Figure 4 - a) Optical image of a small sample area affected by landslides within the AoI; b) the corresponding Log-Ratio layer; c) the output of the segmentation algorithm (obtained using the optimized hr and th values); d) the extracted landslide-related segments (in blue).**



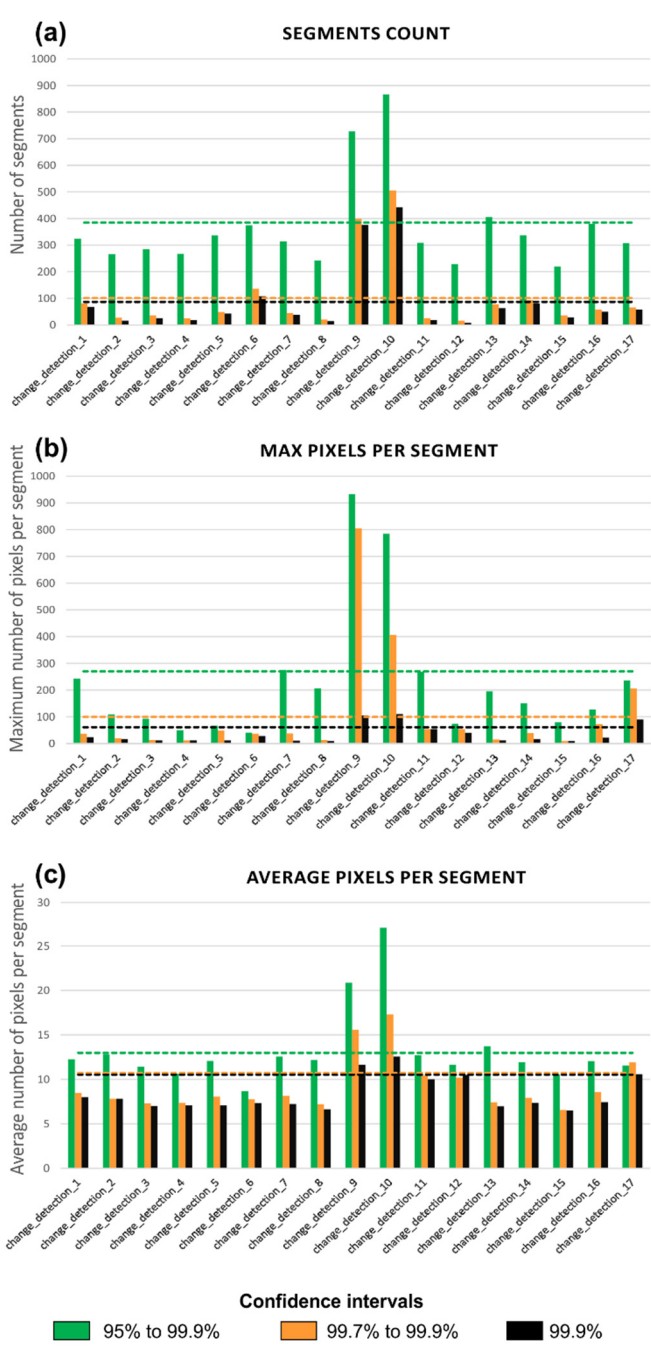


**Figure 5 - Statistics of the segments identified for each change detection. a) number of segments with more than 5 pixels; b-c) maximum and average number of pixels per segment. For the change detections 9 and 10, the two peaks indicate the occurrence of**





**widespread land cover changes. The dashed lines show the 95° percentiles of the distributions (not including change detections 9 and 10).**

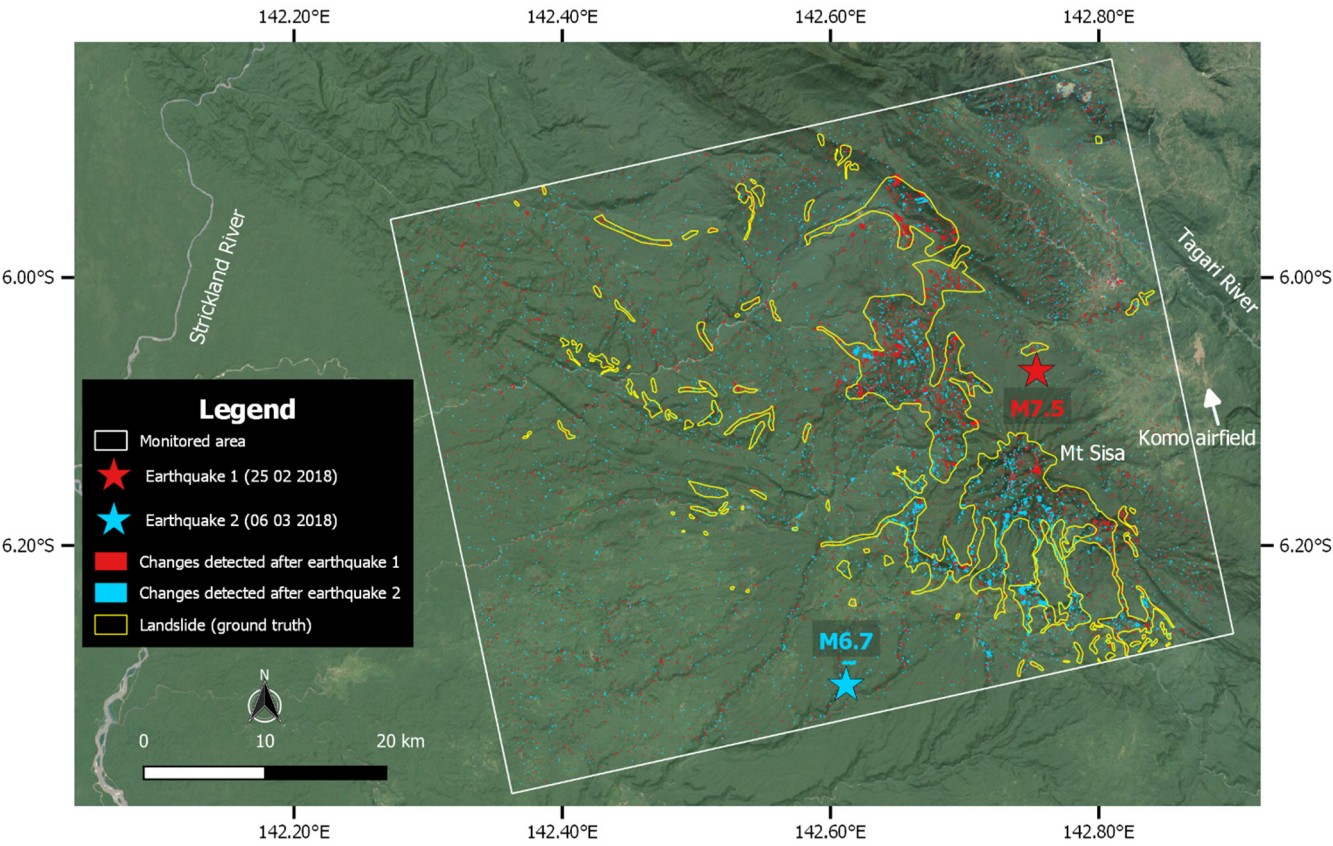


**Figure 6 - The map shows location of the epicenters of the two main earthquakes, and the distribution of segments representing SAR amplitude changes for the change detections 9 and 10. Yellow polygons are areas really affected by landslides. The white rectangle identifies the AoI (see Figure 2).**




**Figure 7 - Comparison of segment's areas statistics related to different change detections.**




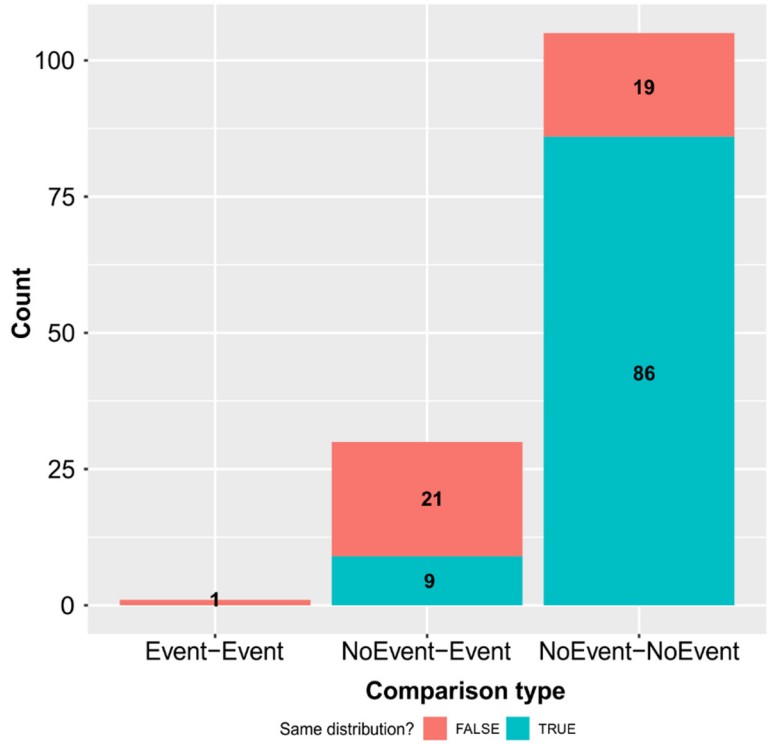

**Figure 8 - Histogram representing the differences, calculated according to the p-value of the Kolmogorov-Smirnov tests, between**

**all the compared distributions.**

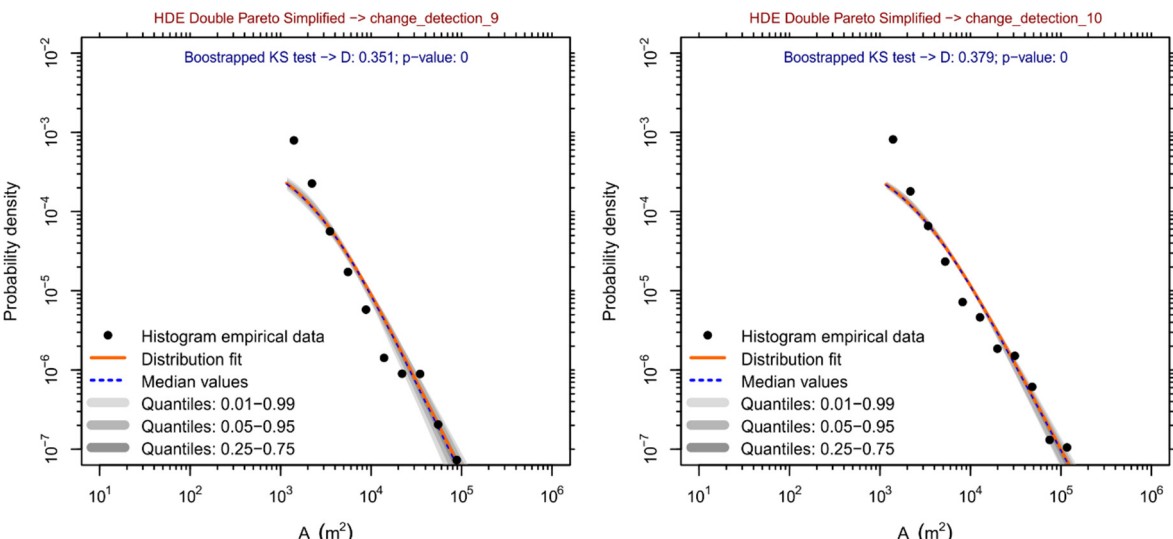

**Figure 9 - Frequency - area distribution of segments resulted in the change detections 9 (on the left) and 10 (on the right), and fitting with a Double Pareto simplified model.**