# Peer review of "A semi-automatic procedure to support the detection of rapid-moving landslides using spaceborne SAR imagery"

_Natural Hazards and Earth System Sciences, 2020_

## Short Comment (SC1) · 10 Mar 2020

**General Comments:**

This research is interested in by readers doing landslides inventory mapping, where SAR intensity images are employed in a large area. This method can overcome the shortage of optical images in case of cloud. The rational and procedure are introduced reasonably. However, some quantitative description of the parameters and the results need be considered carefully. Besides, the current title is somehow inaccurate. The main contribution of the research is the detection of failed landslides (event inventory mapping) rather than rapid moving landslides detection before occurrence. Therefore, I suggest to revise the title.

**Specific Comments:**

(1) Lines 95-96: "Satellites Sentinel-1A and 1B acquire images characterized by a spatial resolution up to 5x5 m, …".
The statement is not correct, the spatial resolutions of Sentinel-1 images are about 5 x 20 m.

(2) Lines 135-136: "…, the resulting stacked images are filtered for speckling reduction using the adaptive Frost filter (Frost et al., 1982), …".
There are many methods to filter speckle noise in SAR images, please give some explanation to use Frost filter in this study.

(3) Lines 146 and 128, the meaning of $\beta_0$ should be unified.

(4) Due to the side-looking imaging geometry of SAR satellites, geometric distortions including layover, shadow and foreshortening are inevitable in mountainous regions, which will cause some blind areas and seriously decrease the capability of landslide detection. In this study, how did the authors deal with geometric distortions during the calculation of SAR amplitude changes?

(5) Line 583: "Flowchart of the automatic steps of the processing chain described in the text."
The authors used the terminology "semi-automatic" in title, however, in here used "automatic". Please unify them. And the manual interaction section should be highlighted.

(6) Figure 2: Please add the coverage of Sentinel-1 SAR images.

(7) Figure 4: (1) Please add a color bar in Figure 4(b) and (c).

(8) Line 290, what do you mean the multiply 196 m2 .5 (980 m2) ?, Combined with the results shown in figure 6, what's the uncertainty and accuracy of the landslides detection? Moreover, what's the minimum area (size) can be detected with SAR intensity change method with high precision?

(9) Figure 6: The obtained results look not good compared with the previous studies (Tessari et al., 2017; Konishi and Suga, 2018) of SAR amplitude images used for landslide detection. Such a result used directly in the detection of landslides will cause serious mis-interpretation. On the other hand, the authors should compare the landslide detection results with the ground truth to evaluate the accuracy and reliability of the method presented in this study, rather than just superimpose the SAR

amplitude changes on the ground truth. Here some quantitative assessments will be better for this method.

(10) Still in Figure 6, the shapes of yellow polygons do not look like landslide, especially the ones close to epicenter of M7.5. So I wonder the surface changes even in the yellow polygons are not landslides but earthquake damage. Can you verify the results?

(11) In general, "rapid-moving landslides" represent the landslides which are deforming with large-gradient without failures so far. Accurately, the landslides detected in this manuscript belong to the event-triggered landslides, i.e. landslides triggered by earthquakes. Please think more about it and make it express more precisely.

---

## Referee Comment (RC1) · Pierluigi Confuorto (Referee) · 13 May 2020

General comments:

The paper entitled "A semi-automatic procedure to support the detection of rapid-moving landslides using space-borne SAR imagery" presents a semi-automatic procedure, exploiting Sentinel-1 SAR images, which evaluates changes of backscattering signals associated to land cover changes due to landsliding.

The manuscript represents a solid and valuable contribution to the current state-of-the-art landslide mapping and detection during in post-emergency phases. The scientific
and the applied methods are excellently depicted and supported by a robust bibliographic background. The results are sound and consistent and supported by a very good statistical analysis, which makes the results very interesting and noteworthy. The discussion of the results in the general framework of the current literature is accurate and addresses all the concerns. The overall quality of the manuscript is very good, with an appropriate number of figures and written in an excellent English, to me.

I have just a few questions which may be addressed in the discussion of the paper and which can be clarified by the authors.

Specific comments:

Title:

The use of rapid landslide in the title can be ambiguous, since the definition of landslide magnitude can be obtained by assessing the intensity or the velocity. It is indeed a movement triggered by sudden events such as earthquakes, however, considering the current timespan between two S1 images. The same aspect should be clarified when using this expression throughout the text. Dataset: I think that more information about the dataset used should be provided. A short reference within the text or by adding a table, along with the frame outline to be inserted in Figure 2, would be more appropriate. Speaking of the dataset, what about the geometry of acquisition? Did you use ascending or descending images? Moreover, do you expect differences in the final results by using both geometries? I also think a short comment on the potential geometric distortions of SAR imagery should be added in the text, if in somehow this may affect the goodness of the results. Structure of the paper: I find ambiguous to write about the results when speaking of the test site. I find more appropriate to separate the result section from the paragraph 3, by adding a fourth paragraph which should address only the results obtained.

Results:

Do you think all the changes detected, even those within the ground truth landslides, can be attributed to landslides? Are there any other land cover changes that can be identified (e.g. deforestation, river deviation, noise, etc.)? You write, indeed (line 301), that many segments outside landslide areas are not attributable to landslides, however, it is possible to find these segments within the ground truth landslides? Do you think is sufficient to discriminate landslide and non-landslide pixels by their number? I think the classification of the detected segments is still a main challenge, which, of course, could be addressed in future work. In this sense, do you think that a validation/comparison with other techniques and other data (e.g. PolSAR, OBIA, InSAR, DTM change detections) may help to better classify land cover changes segments?

Technical comments:

Line 96: please, specify that slightly better than 5 m by 5 m spatial resolution is when dealing with StripMap acquisition mode.

Figure 2: as I said in a previous comment, SAR dataset frame could be added here to have a complete overview of the study area.

Figure 4: please, add a color bar where necessary and the source of the optical image used.

---

## Referee Comment (RC2) · Anonymous Referee #2 · 24 May 2020

General Comments This research is very interested, and I think it represents a valuable contribution to the current state-of-the-art of landslide mapping and detection during post-emergency phases, especially in case of persistent clouds. The Authors apply a change-detection method, classically used in optical remote sensing, to radar images. The rational and methods are well described and presented. I agree with other comments about the title: it is somehow inexact. The main contribution of the research is the detection of earthquake-triggered landslides (event inventory mapping) rather than rapid moving landslides detection before occurrence. Therefore, I agree to revise it. The manuscript is supported by a robust biblio-graphic background. The scientific sound is appropriate and supported by a good statistical analysis, which makes the

results very interesting and noteworthy. The overall quality of the manuscript is very good, with an appropriate number of figures. The English language is good. I have just a few comments as reported in the attached pdf file.

Specific comments - Please provide more information about the used images (ex. image characteristics, geometry of acquisition).

- Please provide more information about georeferencing problems of radar images and associated characteristics that play a role in analysis (i.e. layover, shadow and foreshortening).

- Along the text, it is not clear which processing step is done manually, semi-automatically and in a fully automatic way. Please specify better.

Please also note the supplement to this comment:
https://www.nat-hazards-earth-syst-sci-discuss.net/nhess-2020-55/nhess-2020-55-RC2-supplement.pdf

**Supplement:**

**A semi-automatic procedure to support the detection of rapidmoving landslides using spaceborne SAR imagery**

Giuseppe Esposito1, Ivan Marchesini2, Alessandro Cesare Mondini2, Paola Reichenbach2, Mauro Rossi2, Simone Sterlacchini3

[revised manuscript text omitted]

---

## Author Comment (AC1) · 3 Jul 2020

General comments:

The paper entitled "A semi-automatic procedure to support the detection of rapid-moving landslides using space-borne SAR imagery" presents a semi-automatic procedure, exploiting Sentinel-1 SAR images, which evaluates changes of backscattering signals associated to land cover changes due to landsliding. The manuscript represents a solid and valuable contribution to the current state-of-the art landslide mapping and detection during in post-emergency phases. The scientific and the applied methods are excellently depicted and supported by a robust bibliographic background. The

results are sound and consistent and supported by a very good statistical analysis, which makes the results very interesting and noteworthy. The discussion of the results in the general framework of the current literature is accurate and addresses all the concerns. The overall quality of the manuscript is very good, with an appropriate number of figures and written in an excellent English, to me. I have just a few questions which may be addressed in the discussion of the paper and which can be clarified by the authors.

Author response: We are grateful to the Reviewer for the valuable comments and suggestions relevant for the improvement of the manuscript. Point-by-point responses to all comments are outlined below. The proposed changes to the text and Figures are provided in the attached pdf file.

Specific comments

Title: The use of rapid landslide in the title can be ambiguous, since the definition of landslide magnitude can be obtained by assessing the intensity or the velocity. It is indeed a movement triggered by sudden events such as earthquakes, however, considering the current timespan between two S1 images. The same aspect should be clarified when using this expression throughout the text.

Author response: We agree with this comment. Both the title and some sentences in the text have been modified accordingly. In addition, this aspect has been explained into the Introduction section, by specifying that we focus on rapid-moving landslides since they determine more evident land cover changes with respect to slow-moving mass movements.

Dataset: I think that more information about the dataset used should be provided. A short reference within the text or by adding a table, along with the frame outline to be inserted in Figure 2, would be more appropriate.

Author response: Information on the dataset has been provided both into the sections

2.1 and 3. Further specifications have been inserted in section 3, as highlighted below:

"Considering that the majority of the slopes in the study area are exposed towards West, to limit geometrical distortions in the single images and in the change detection estimation, we preferred to use IW-SLC products acquired in ascending mode, with a VV-VH polarization. Each IW product is collected with a swath characterized by a width of 250 km, subdivided in turn to three sub-swaths containing one image per polarization consisting of a series of bursts which are processed as independent SLC images."

A new version of Figure 2, including the spatial extent of the used Sentinel-1 SAR images, has been also prepared, as shown in the attached pdf file.

Speaking of the dataset, what about the geometry of acquisition? Did you use ascending or descending images? Moreover, do you expect differences in the final results by using both geometries? I also think a short comment on the potential geometric distortions of SAR imagery should be added in the text, if in somehow this may affect the goodness of the results.

Author response: In chapter 3 we have specified that this first application of the processing chain is based only on images acquired in ascending mode. This choice was taken a priori, taking into account that most of the slopes in the study area are exposed towards West, with the aim of limiting the inclusion of geometrical distortions. Besides this, we appreciate this valuable comment that gives us a starting point for improving the proposed procedure and considering the descending images in future steps. We agree that images acquired in both geometries can improve the quality of results, especially in mountainous areas. However, in this first application we focused mainly on the implementation of the processing chain, trying to get reasonable results in terms of output to a real case study. In the next steps, we will work to improve both detection and localization of landslide-related land cover changes, taking into account the possibility of combining ascending and descending images. We have added short comments in the Discussion section, as requested by the Reviewer.

Structure of the paper: I find ambiguous to write about the results when speaking of the test site. I find more appropriate to separate the result section from the paragraph 3, by adding a fourth paragraph which should address only the results obtained.

Author response: We structured the paper by describing firstly the implemented procedure, and subsequently the application in Papua New Guinea. The latter is described within the chapter 3 that is thus aimed at outlining the results obtained with the procedure described before. For this reason, we propose to do not change the structure of the paper.

Results: Do you think all the changes detected, even those within the ground truth landslides, can be attributed to landslides? Are there any other land cover changes that can be identified (e.g. deforestation, river deviation, noise, etc.)? You write, indeed (line 301), that many segments outside landslide areas are not attributable to landslides, however, it is possible to find these segments within the ground truth landslides? Do you think is sufficient to discriminate landslide and non-landslide pixels by their number?

Author response: Thanks for this comment that allows us to clarify this crucial point of the paper. It is possible to find segments related to noise and local stream changes also within the yellow polygons, where landslides occurred. However, segments related to these type of changes are relatively smaller than those ascribable to landslides, as shown in the unchanged zones represented in Figure 4(d). The occurrence of many and large landslide-related segments has strongly influenced the statistics of change detections 9 and 10 shown in Figure 5, with respect to the other change detections characterized mostly by noise-related smaller segments. Therefore, both the size and the number of segments resulted discriminant for landslides detection. We exclude changes related to earthquake damage into the yellow polygons, given that the study area is sparsely populated.

I think the classification of the detected segments is still a main challenge, which, of

course, could be addressed in future work. In this sense, do you think that a validation/comparison with other techniques and other data (e.g. PolSAR, OBIA, InSAR, DTM change detections) may help to better classify land cover changes segments?

Author response: We agree with the Reviewer. A future integration of different types of data and techniques may be useful to better classify the land cover changes and improve the landslides detection. Data at higher spatial and temporal resolution providing different types of information, for example, may allow validating and/or improving the current outcomes. However, it is worth noting that one of the strengths of our procedure is the use of data that are freely accessible, and with a constant revisiting time. This supported us in implementing an automatic processing chain.

Technical comments:

Line 96: please, specify that slightly better than 5 m by 5 m spatial resolution is when dealing with StripMap acquisition mode.

Author response: We agree with this comment. The sentence has been modified as follows:

"Satellites Sentinel-1A and 1B acquire images characterized by pixels with sizes ranging from 5 (range) $\times$ 20 (azimuth) m in the default acquisition mode for land observations (Interferometric Wide Swath mode - IW) up to 5x5 m, in the Strip Map mode".

Figure 2: as I said in a previous comment, SAR dataset frame could be added here to have a complete overview of the study area.

Author response: We agree with this comment and a new version of Figure 2, including the spatial coverage of the used Sentinel-1 SAR images, has been prepared. Please, see the attached pdf file.

Figure 4: please, add a color bar where necessary and the source of the optical image used.

Author response: We added a color bar only in the Figure 4(b), since it was missing in the early version. Further comments aimed to clarify the contents reported in the Figure 4(c), as well as the source of optical images used in the Figures 4(a) and 4(d) are added into the caption. Please, see the attached pdf file.

Please also note the supplement to this comment:
https://www.nat-hazards-earth-syst-sci-discuss.net/nhess-2020-55/nhess-2020-55-AC1-supplement.pdf

---

## Author Comment (AC2) · 3 Jul 2020

General comments:

This research is very interested, and I think it represents a valuable contribution to the current state-of-the-art of landslide mapping and detection during post-emergency phases, especially in case of persistent clouds. The Authors apply a change-detection method, classically used in optical remote sensing, to radar images. The rational and methods are well described and presented. I agree with other comments about the title: it is somehow inexact. The main contribution of the research is the detection of earthquake-triggered landslides (event inventory mapping) rather than rapid moving

landslides detection before occurrence. Therefore, I agree to revise it. The manuscript is supported by a robust biblio-graphic background. The scientific sound is appropriate and supported by a good statistical analysis, which makes the results very interesting and noteworthy. The overall quality of the manuscript is very good, with an appropriate number of figures. The English language is good. I have just a few comments as reported in the attached pdf file.

Author response: We are grateful to the Reviewer for the appreciated comments and suggestions aimed at improving our manuscript. We agree with the previous comment. As highlighted in the attached pdf file, both the title and other issues identified by the Reviewer throughout the manuscript have been revised accordingly.

Specific comments

Please provide more information about the used images (ex. Image characteristics, geometry of acquisition).

Author response: Information on the dataset has been provided both into the sections 2.1 and 3. Further specifications have been inserted in section 3, as highlighted below:

"Considering that the majority of the slopes in the study area are exposed towards West, to limit geometrical distortions in the single images and in the change detection estimation, we preferred to use IW-SLC products acquired in ascending mode, with a VV-VH polarization. Each IW product is collected with a swath characterized by a width of 250 km, subdivided in turn to three sub-swaths containing one image per polarization consisting of a series of bursts which are processed as independent SLC images."

A new version of Figure 2, including the spatial extent of the used Sentinel-1 SAR images, has been also prepared, as shown in the attached pdf file.

Please provide more information about georeferencing problems of radar images and associated characteristics that play a role in analysis (i.e. layover, shadow and fore-shortening).

Author response: More information about this point are provided both in the Introduction and Discussion sections, as highlighted below:

Introduction: "geometric distortions, such as layover and shadowing due to the side-looking acquisition geometry of SAR sensors, that can affect the quality of the images over mountainous areas, where landslides are likely to occur."

Discussion: "Another improvement may consist in the use of images acquired in both ascending and descending geometries. The use of ascending images was only related to focus this first step of the work on the implementation of the entire processing chain, that we tried to simplify as much as possible. In fact, there is no doubt that combining images acquired in ascending and descending geometries can improve the quality of results, representing a non-trivial advancement of the procedure that was out of the aim of this first implementation. The a priori choice of using ascending products was based on the findings that most of the slopes in the study area are exposed towards West, with the aim of limiting the inclusion of geometrical distortions in the change detection products."

Along the text, it is not clear which processing step is done manually, semi-automatically and in a fully automatic way. Please specify better.

Author response: Considering this comment, we probably have improperly termed the proposed procedure as "semi-automatic". In fact, the operations described in the flowchart run in an automatic way but they need a one-time calibration phase to define both values of the parameters required for the segmentation and some statistics. Therefore, we preferred to delete "semi-automatic" from the title and within the revised version of the manuscript. Moreover, it is worth noting that information on how we automatized the described procedure is provided in the paragraph 2.5, that we have renamed as follows: "Automatic implementation of the processing chain".

Please also note the supplement to this comment:

https://www.nat-hazards-earth-syst-sci-discuss.net/nhess-2020-55/nhess-2020-55-AC2-supplement.pdf

---

## Author Comment (AC3) · 3 Jul 2020

General comments:

This research is interested in by readers doing landslides inventory mapping, where SAR intensity images are employed in a large area. This method can overcome the shortage of optical images in case of cloud. The rational and procedure are introduced reasonably. However, some quantitative description of the parameters and the results need be considered carefully. Besides, the current title is somehow inaccurate. The main contribution of the research is the detection of failed landslides (event inventory mapping) rather than rapid moving landslides detection before occurrence. Therefore,

[Figure]

I suggest to revise the title.

Author response: We agree with this comment. The title has been modified accordingly, by deleting the term "rapid-moving" as indicated below:

"A spaceborne SAR-based procedure to support the detection of landslides"

Specific comments:

(1) Lines 95-96: "Satellites Sentinel-1A and 1B acquire images characterized by a spatial resolution up to 5x5 m, . . .". The statement is not correct, the spatial resolutions of Sentinel-1 images are about 5 x 20 m.

Author response: This point has been clarified in the text as indicated below:

"Satellites Sentinel-1A and 1B acquire images characterized by pixels with sizes ranging from 5 (range) $\times$ 20 (azimuth) m in the default acquisition mode for land observations (Interferometric Wide Swath mode - IW), up to 5x5 m in the Strip Map mode".

(2) Lines 135-136: ". . ., the resulting stacked images are filtered for speckling reduction using the adaptive Frost filter (Frost et al., 1982), . . .". There are many methods to filter speckle noise in SAR images, please give some explanation to use Frost filter in this study.

Author response: We agree with the Reviewer. We chose the Frost filter following the results of some previous studies. In particular, according to Schellenberger et al. (2012), it is one of the best choices in mountainous environments, it can account for the local properties of the terrain backscatter (and landslides are local objects in this context), and it was already used successfully in previous studies dealing with landslides (Mondini, 2017). We acknowledge that using different filters we might have obtained slightly different results, and this is now discussed.

(3) Lines 146 and 128, the meaning of $\beta 0$ should be unified.

Author response: We agree with this comment. Appropriate corrections have been

done accordingly throughout the text.

(4) Due to the side-looking imaging geometry of SAR satellites, geometric distortions including layover, shadow and foreshortening are inevitable in mountainous regions, which will cause some blind areas and seriously decrease the capability of landslide detection. In this study, how did the authors deal with geometric distortions during the calculation of SAR amplitude changes?

Author response: Pixels in layover and shadows were obtained using the "SAR simulation Terrain Correction" tool available in SNAP, exploiting the SRTM 1Sec DEM, and then masked before running the statistical analysis. Foreshortening was partially mitigated by means of the reprojection procedure. We verified that the amount of the study area affected by such distortions is less than 1%.

(5) Line 583: "Flowchart of the automatic steps of the processing chain described in the text." The authors used the terminology "semi-automatic" in title, however, in here used "automatic". Please unify them. And the manual interaction section should be highlighted.

Author response: Considering this comment, we probably have improperly termed the proposed procedure as "semi-automatic". In fact, the operations described in the flowchart run in an automatic way but they need a one-time calibration phase to define both values of the parameters required for the segmentation and some statistics. Therefore, we preferred to delete "semi-automatic" from the title and within the revised version of the manuscript. Moreover, it is worth noting that information on how we automatized the described procedure is provided in the paragraph 2.5, that we have renamed as follows: "Automatic implementation of the processing chain".

(6) Figure 2: Please add the coverage of Sentinel-1 SAR images.

Author response: We accept this comment. A new version of Figure 2, including the spatial coverage of the used Sentinel-1 SAR images, has been prepared and shown in

the attached pdf file.

(7) Figure 4: (1) Please add a color bar in Figure 4(b) and (c).

Author response: We added a color bar only in the Figure 4(b), since it was missing in the early version. Further comments aimed to clarify the contents reported in the Figure 4(c), as well as the source of optical images used in the Figures 4(a) and 4(d) are added into the caption. Please, see the attached pdf file.

(8) Line 290, what do you mean the multiply 196 m2 .5 (980 m2)?, Combined with the results shown in figure 6, what's the uncertainty and accuracy of the landslides detection? Moreover, what's the minimum area (size) can be detected with SAR intensity change method with high precision?

Author response: 980 m2 derives from the product of a single pixel area, roughly equal to 196 m2 (14x14 m2 considering that 14 m is the Log-Ratio pixel size calculated after the multi-looking process), times 5 that is the minimum number of pixels included within a segment. We decided to use 5 pixels after a general evaluation of the preliminary landslide-related images published on news websites and social networks, and considering that the detection of smaller segments in the test area was not significant at the scale of our analysis. Therefore, 980 m2 is a minimum area that we retained as potentially affected by a landslide. Moreover, our procedure is not aimed at landslide mapping but at a preliminary detection and rough localization of landslides, considering as minimum area affected by landslides the one selected according to the decided pixel threshold only.

(9) Figure 6: The obtained results look not good compared with the previous studies (Tessari et al., 2017; Konishi and Suga, 2018) of SAR amplitude images used for landslide detection. Such a result used directly in the detection of landslides will cause serious mis-interpretation. On the other hand, the authors should compare the landslide detection results with the ground truth to evaluate the accuracy and reliability of the method presented in this study, rather than just superimpose the SAR amplitude

changes on the ground truth. Here some quantitative assessments will be better for this method.

Author response: Thank you for this comment that gives us the opportunity to explain better a relevant point of our work. Both the cited studies were based on X-band SAR data acquired at high resolution and focused on areas smaller than the one analyzed in our study. This allowed both detection and mapping operations with a relatively high accuracy. In addition, both studies refer to geographic areas with different geological, geomorphological and land use properties with respect to the one analyzed in this work, which are also exposed to different landslide typologies. In the light of this, we believe that suitable comparisons should be possible if the same data were applied in the same area with similar techniques. Besides this, we would highlight that we present an attempt that use freely accessible C-band data, exploiting their constant availability with respect to other SAR products. The aim of the processing chain is in fact the early detection and localization of land cover changes induced by landslides over wide areas (i.e. thousands of square kilometers). The Figure 6 shows that the calculated segments concentrate mostly in the yellow polygons, where numerous landslides really occurred in the field. Considering this a first test, we retain the outcome satisfactory. Further detailed analyses, aimed at reducing some limitations of the used data, should be done for future improvements of the processing chain.

(10) Still in Figure 6, the shapes of yellow polygons do not look like landslide, especially the ones close to epicenter of M7.5. So I wonder the surface changes even in the yellow polygons are not landslides but earthquake damage. Can you verify the results?

Author response: The yellow polygons in Figure 6 (see legend) highlight the areas affected by landslides. The polygons were drawn independently from the segmentation, by means of a rough interpretation of optical data, with the aim of delimiting areas where landslides occurred in the field. In the test area, we did not perform a detailed mapping since we consider it out of the aims of the study. We used the yellow polygons to check whether the segments (red and blue pixels in Figure 6) obtained with our

procedure were located in areas where the concentration of landslides was high and evident. We exclude earthquake damage into the yellow polygons, given that the study area is sparsely populated.

(11) In general, "rapid-moving landslides" represent the landslides which are deforming with large gradient without failures so far. Accurately, the landslides detected in this manuscript belong to the event-triggered landslides, i.e. landslides triggered by earthquakes. Please think more about it and make it express more precisely.

Author response: We agree with this comment. The title and the text have been modified accordingly.

Please also note the supplement to this comment:
https://www.nat-hazards-earth-syst-sci-discuss.net/nhess-2020-55/nhess-2020-55-AC3-supplement.pdf

**Supplement:**

**A  spaceborne SAR-based procedure to support the detection of  landslides**

[revised manuscript text omitted]
; 2) geometric distortions, the such as layover and shadowing due to the side-looking acquisition geometry of SAR sensors, that can affect the quality of the images over mountainous areas, where landslides are likely to occur; 3) the difficulty in using the SAR signal in traditional statistical classification approaches mainly due to speckling. A successful example of the use of amplitude variations of the radar signal to analyze landslides is described by Zhao et al. (2013), which inferred the Jiweishan rock slide in China using changes in SAR backscattering intensity in ALOS/PALSAR images. Tessari et al. (2017) verified that when the phase information cannot be exploited, amplitude of the reflected signal is very useful to detect and map rapid-moving landslides that cause significant variations in the ground morphology and land cover. Mondini (2017) proved that both landslides and flooded areas can be detected by verifying changes in the spatial autocorrelation in a multi-temporal series of SAR images. Konishi and Suga (2018) also identified a series of landslides in Japan by analyzing intensity correlation between pre- and post-event SAR images.

Besides the described techniques, recent advances in SAR technology are promoting the use of polarimetric SAR data (PolSAR) characterized by full-polarimetric information (i.e., acquired in single polarization, dual polarization, and fully polarimetric modes) for a target in the form of the scattering matrix (Skriver, 2012). According to Plank et al. (2016), these data provide more information on the ground, which enables a better land cover classification and landslide mapping. Successful applications were described by Yamaguchi (2012), Shimada et al. (2014), Li et al. (2014) and Plank et al. (2016).

The use of SAR data to analyze landslides and/or potentially unstable slopes should hence increase, also in relation to a series of valuable technical innovations. The improved revisiting times and spatial resolution of the images, for example, represent a key factor during disaster response operations, when a preliminary localization of areas potentially affected by major landslides is crucial. Revisiting times have been in fact reduced from 35 days of ERS and Envisat satellites, to 12 hours (at 40°latitude, in case of emergency response) of the COSMO-SkyMed constellation (Casagli et al., 2017). The enhanced spatial resolution (azimuth or along-track resolution x range or across-track resolution) of images spans in the order of few meters (i.e. 1-10 m), resulting more detailed with respect to the coarser resolution of the first-generation satellites characterized by pixel sizes of 10-30 up to 100 meters (Plank, 2014).

Among the most advanced SAR spaceborne systems (Casagli et al., 2017), there are those of the mission Sentinel-1 operated by the European Space Agency (ESA) in the frame of the European Union's Copernicus Programme. Satellites Sentinel-1A and 1B acquire images characterized by a spatial resolution pixels with sizes ranging from 5 (range) × 20 (azimuth) m in the default acquisition mode for land observations (Interferometric Wide Swath mode - IW), up to 5x5 m, depending on the in

[revised manuscript text omitted]

235    close to zero.

In this way, all the segments of the "average layer" characterized by $\mu_s$ values larger than $|\mu+(2\bar{\sigma})|$ are then extracted and classified. Segments with $\mu_s$ values lower than a confidence interval of 95% ($\mu_s<|\mu+(2\bar{\sigma})|$) are instead discarded. Segments where $\mu_s$ is greater than $|\mu+(2\bar{\sigma})|$ and smaller than $|\mu+(3\bar{\sigma})|$ are reclassified to the integer value of 2. Similarly, the values 3 and 4 are used to classify segments with $\mu_s$ values included in the range $|\mu+(3\bar{\sigma})|$ to $|\mu+(4\bar{\sigma})|$, and larger than $|\mu+(4\bar{\sigma})|$,

240    respectively. All these segments form a new raster layer representing a map of areas characterized by relevant SAR amplitude changes including those affected by rapid slope movements.

In order to refine this map, all the segments with the same values (i.e., 2, 3, or 4), that are spatially contiguous and are formed by at least a user-defined minimum arbitrary size in terms of pixels (i.e. minimum detectable landslide area) are merged together, and the following statistics are then computed: 1) count of merged segments; 2) maximum number of

245    pixels included within a single segment; and 3) average number of pixels included within a single segment.

The final segment map produced by the  processing chain is georeferenced in the WGS84 reference system (EPSG 4326) by means of the Terrain correction tool of SNAP.

**2.5  Automatic implementation of the processing chain**

The processes described before have been implemented in two groups of scripts that can be executed automatically (in-chain), according to the flowchart shown in Figure 1. They are run after a preliminary one-time calibration phase, operated manually by the user and consisting of: 1) the tuning of the i.segment parameters, carried out with the expert-based segmentation of an event-related LR image (paragraph 2.3); 2) the computation of reference $\mu$ and $\bar{\sigma}$ related to no-change conditions, as described in the paragraph 2.4.

The python-based script (Fig. 1, Data ingestion) is devoted to the automatic querying and downloading of Sentinel-1 SAR images from the ESA Sentinel Data Hub. The script, based on the SentinelSat toolbox (Kersten et al., 2018), is set to query the Sentinel Data Hub with a daily frequency even though new images may be available every 6 or 12 days, depending on the geographic area.

The group of scripts, written in GNU/Bash programming language (Fig. 1), is aimed at: (i) pre-processing the Sentinel-1 images (section 2.1), (ii) detection of the changes in SAR amplitude and production of Log-Ratio maps (section 2.2), (iii) segmentation of the LR maps (section 2.3) and, (iv) identification of areas potentially affected by land cover changes (section 2.4). This group of scripts is executed automatically when new Sentinel-1 images are available and downloaded by the python-based script.

The bash-scripts require the following settings defined by the user: 1) the path of the folder where the downloaded SAR images are stored; 2) values of the parameters required for the segmentation (see section 2.3), and (3) the spatial coordinates of the area of interest (if it is a portion of the downloaded SAR images). No further information is needed since the commands are executed in a unique automatic sequence. To survey the same area for an unlimited time period, all these settings have to be defined only for the chain initialization.

**3 The Papua New Guinea test site**

We selected as test site, an area located in central Papua New Guinea (Fig. 2) that was affected by a severe seismic sequence at the beginning of 2018. On 25 February, the area was hit by a main seismic event (M7.5) followed by several aftershocks, including a M6.7 earthquake on 6 March. The strong mainshock, rather superficial with a hypocentral depth at 23.4 km (USGS, 2018), caused building collapses, road damage and widespread landslides mostly along the Tagari river valley and

the slopes of Mount Sisa (McCue et al., 2018). According to the International Federation of Red Cross and Red Crescent Societies (IFRC, 2018), more than 100 people died, most of them due to landslides.

To test the implemented procedure, we  analyzed an area of about 3000 km$^2$ in the mountainous region close to the epicenters of the mainshock (AoI in Fig. 2), where preliminary information on landslides were available (Petley, 2018a, b).

280 To simulate a periodic survey covering pre- and post-earthquake periods, we downloaded 36 Sentinel-1 images from the Sentinel Data Hub (https://scihub.copernicus.eu/) acquired along the satellite track n.82  with a temporal frequency of 12 days, from 12 November 2017 to 6 June 2018. Considering that the majority of the slopes in the study area are exposed towards West, to limit geometrical distortions in the single images and in the change detection estimation, we preferred to use IW-SLC products acquired in ascending mode, with a VV-VH polarization. Each IW product is collected

285 with a swath characterized by a width of 250 km, subdivided in turn to three sub-swaths containing one image per polarization consisting of a series of bursts which are processed as independent SLC images.

The downloaded images were used to perform a total of 17 change detection analyses which resulted in likewise LR layers, with a pixel size of about 14 m. The values of the segmentation parameters were defined with an interactive manual analysis (see section 2.3) by segmenting the "pre-post M7.5 earthquake" LR layer, selecting the spatial kernel size (*hs*) of 10 pixels

290 (see section 2.3), and setting the maximum number of iterations to 200. This size of the spatial kernel was set to 10 pixels to detect significant differences of LR values during the smoothing stage of the segmentation process, taking into account the approximate expected size of the land cover changes. In the interactive (manual) analysis, we selected bandwidth sizes (*hr*, see section 2.3) ranging from 0.0005 to 0.016, and thresholds (*th*) from 0.001 to 0.016 (Fig. 3), obtaining 20 different parameter combinations. For each couple of parameters, the number of generated segments and their average size were

295 plotted  in Fig. ure 3. Points highlight the major impact of the *hr* parameter with respect to the role played by the threshold (*th*) parameter, in defining the number of total generated segments. Below an *hr* value of 0.004 over segmentation occur, whereas for *hr* values equal or larger than 0.004, the number of generated segments tends to become small and constant. With the aim of avoiding over segmentation while maintaining a reasonable average size of the segments (to be able to delineate also small patches of the terrain where changes occurred), and considering a visual inspection of the

300 segmentation results, we decided to run i.segment in the automatic processing chain using the following set of parameters values: *hs*=10, *hr*=0.004, *th*=0.008, *minsize*=2, *iterations*=200 (see section 2.3).

After the segmentation of the 17 LR layers, areas affected by layover and shadowing effects were masked out in order to avoid errors in the statistical analysis described below and in the localization of potential landslides. The mask was developed in SNAP by means of the SAR Simulation Terrain Correction tool, exploiting the SRTM 1Sec DEM.

305 The segments with a minimum size of 5 pixels were extracted in the area of interest (an example is shown in Fig. 4d), and statistics were calculated according to the confidence intervals described in the methodology section. We decided to select only the segments with a minimum size of 5 pixels, corresponding to a minimum area of about 980 m$^2$ (i.e. a single-pixel area roughly equal to 196 m$^2$ times 5 pixels), after a  general evaluation of the preliminary landslide-related images published on news websites and social networks, and considering that the  detection of smaller

310 segments in the test area  was not significant at the scale of our analysis for detecting landslides. It is worth noting that accuracy of such a minimum area is not accurate due to the use of a geographic (not projected) reference system (WGS84).

In Fig.ure 5, statistics of the selected segments are displayed for each change detection. The analysis of the histograms revealed  two main peaks  corresponding to the change detections 9 and 10. Change detection 9 considers
315 images acquired before and after the M7.5 earthquake, whereas change detection 10 the images acquired on 28 February and 12 March 2018. The first peak highlights widespread changes related also to landslides extensively documented after the M7.5 event (Petley, 2018a). The second peak was instead unexpected and was probably due to the occurrence of further landslides triggered by the M6.7 event on 6 March 2018. In Fig.ure 6, segments related to these two peaks are displayed (red pixels = change detection 9; blue pixels = change detection 10). To check whether these segments were effectively located in
320 areas characterized by a high concentration of seismic-induced landslides, we analyzed optical images available on the Planet explorer application (Planet, 2017). By means of a visual interpretation, we identified the zones (the yellow polygons shown in Fig. 6) where clusters of landslides occurred, verifying a general accordance with the spatial distribution of both red and blue segments.
325

[revised manuscript text omitted]

365 **4 Discussion**

In this article, we describe a  processing chain aimed at identifying SAR amplitude changes that can be partially explained by the occurrence of  mass movements. We have selected SAR data since they have the advantage to be not affected by the cloud cover disturbance. In fact, as described by Mondini et al. (2019), the use of SAR amplitude data can mitigate the cloud coverage issue and can allow detecting landslides that, otherwise, might remain unknown or
370 unnoticed for a long time. In this way, the procedure can be exploited for a "continuous", in terms of time, slope monitoring activity, even if failures occur during long-lasting periods of precipitation and persistent cloud cover that do not allow to use optical data for a rapid and detailed landslide recognition. In the selected study area, a widespread cloud cover persisted for several weeks during and after the seismic sequence. The first cloudless optical image of the area damaged by the seismic shaking was published by the daily monitoring service delivered by ©2019 Planet Labs Inc. (www.planet.com) on 25 March,

almost one month after the M7.5 mainshock that triggered numerous landslides. The high cloud persistence is quite common in Papua New Guinea, and in fact this is included in the cloudiest regions of the world with annual cloud frequency (proportion of days with a positive cloud flag) higher than eighty percent (Wilson and Jets, 2016; Mondini et al., 2019). As consequence, the use of optical data in this area, and in other mountainous regions exposed to prolonged rainfall related to monsoons, cyclones or other persistent meteorological systems results tricky.

The obtained results depend on the definition of the image pre-processing and segmentation parameters that should be calibrated a priori (see sections 2.3 and 2.4). While images geometric and radiometric corrections are quite standard and well-accepted procedures, the SAR multiplicative noise filtering remains a largely discussed point in the scientific literature and there is not a consensus on the selection of strategies. We choose the Frost filter because it already proved to be properly working in mountainous environments (Schellenberg et al., 2012) and it was used successfully in previous studies dealing with landslides (Mondini, 2017). We acknowledge that different filters might have brought different results or requested a different tuning of the segmentation procedure. The impact of different filters on our procedure might be an interesting follow up of this work. Another improvement may consist in the use of images acquired in both ascending and descending geometries. The use of ascending images was only related to focus this first step of the work on the implementation of the entire processing chain, that we tried to simplify as much as possible. In fact, there is no doubt that combining images acquired in ascending and descending geometries can improve the quality of results, representing a non-trivial advancement of the procedure that was out of the aim of this first implementation. The a priori choice of using ascending products was based on the findings that most of the slopes in the study area are exposed towards West, with the aim of limiting the inclusion of geometrical distortions in the change detection products.

The tuning of the segmentation parameters is the key element for identifying areas affected by significant land cover changes, also induced by rapid-moving slope movements. This process can be retained event-dependent, requiring a well-known landslide event occurred in the past in the analyzed area or in zones with similar topographic and land use characteristics. In the case study here described, definite values of the segmentation parameters were obtained by segmenting the pre-post M7.5 earthquake LR layer, by testing different values combinations (Fig. 3). This may represent a limit of the proposed procedure if one would apply it in different geomorphic settings without past landslide events, or identifying different types of slope failures. On the other hand, if a proper event-based tuning operation is performed, a continuous monitoring of slopes can be efficiently  carried out without temporal limitations, exploiting both pre- and post-event available images, as done in the current case history. The described application highlighted in fact that by keeping the same parameters values, landslides and other land cover changes triggered by the M6.7 aftershock were also detected. The occurrence and location of these secondary failures (blue pixels in Fig. 6) were not known before our analysis because not reported by news and local government websites, and also missing in the maps of the Copernicus Emergency Management Service (https://emergency.copernicus.eu/mapping/list-of-components/EMSR270) activated for the disaster response. The general lack of information related to these failures was likely due to a series of issues affecting both the field and the satellite surveys in the aftermath of the M6.7 earthquake. In fact, an effective assessment in the field was impeded by the

road damages caused by the mass movements triggered by the previous major M7.5 event, whereas the use of optical satellite images was hampered by a widespread cloud cover that, as stated before, persisted during several weeks after the two main seismic shaking. The first information about the occurrence of these landslides were provided online by Petley (2018b), about one month later, without a clear indication of their relationships with the M6.7 earthquake. The detection of this second set of failures in areas poorly affected by slope movements triggered by the M7.5 event demonstrates the relevant usefulness of the proposed processing chain.

The segments located mostly outside the landslide affected areas affected by landslides are caused by other land cover changes that are out of the aims of this study, or by random noise effects. Segments related to these changes can be easily identified because composed by an average number of pixels close to ten, as detected in all the change detections, whereas segments related to landslides (i.e. change detections 9 and 10) are characterized by a higher number of pixels (Fig. 5c).

A suitable segmentation can allow hence to get statistical evidences of event landslides occurrence. Statistical distributions of the three parameters shown in Fig. ure 5 provide distinctive signatures of widespread land cover changes triggered by the M7.5 mainshock and by the M6.7 aftershock. It is worth noting, however, that 95-percentiles highlighted in the plots are exceeded also by other peaks (e.g., change detection 13 in the segment count), that cannot be considered as diagnostic of landslide occurrence since they are ephemeral, and are not steady in all the three plots as the change detections 9 and 10. In case of small-scale landslides occurring in localized portions of a wide area, the related statistical signals may result imperceptible if these are of the same magnitude of other previous and successive signals not related to landslides. In cases like this, distinctive evidence of slope failures can be achieved by starting the process chain with a smaller subset of the LR layer (i.e. monitored area).

Overlapping between the segments (i.e. change detections 9 and 10) to ground truth data revealed that largest SAR amplitude changes correspond often with landslides (Fig. 6). A further evidence was provided by the statistical distributions shown in Fig. ure 9, resulted similar to those estimated by other landslide-related studies (Stark and Hovius, 2001; Malamud et al., 2004; Rossi et al., 2012; Schlögel et al., 2015).

The outcomes of this study represent a concrete example on how to exploit the relevant advantages of Open Source software with a command line interface (i.e. SNAP and GRASS GIS) to implement automatic processing chains. Moreover, it is worth noting that the proposed methodology can be properly adopted to monitor areas in the order of thousands of square kilometers if powerful hardware resources are available. In fact, the pre-processing and segmentation steps require significant amounts of calculation power and memory. It is well known that the Mean Shift is a time-consuming algorithm for large datasets (Wu and Yang, 2007), and convergence for large areas can be reached in dozens of hours. Segmentation times are proportional to the dimensions of the monitored area, and to the selected spatial kernel size ($hs$).

A final remark concerns the occurrence of landslides in the study area. Generally, landslides in the mountainous sectors of Papua New Guinea are very common processes. Earthquakes with a magnitude greater than 5 are among the dominant factors triggering widespread landslides. According to Robbins and Petterson (2015), such earthquakes occur regularly in the country but records of the triggered landslides are surprisingly lacking. The lack of systematic reporting and the remoteness

of communities affected by such events, also impeded an adequate characterization of landslide hazard and risk (Blong, 1986). Robbins et al. (2013) stated that landslides occur annually, and failures tend to range from few cubic meters of material to mass movements with estimated volumes of $1.8 \times 10^9$ m$^3$, varying from debris slides, avalanches and flows to translational and rotational slides. In this framework, the landslide detection procedure described in the article may result a relevant tool for local authorities of countries characterized by extensive remote areas repeatedly affected by slope failures, and for the humanitarian organizations operating in response to geo-hydrological disasters.

**5 Conclusions**

This study presented a  procedure aimed to support the detection of  landslides inducing sharp land cover changes on vast mountainous areas. It is based on SAR data acquired systematically by the Sentinel-1 satellites. The computation of the Log-Ratio index and segmentation of the consequent raster layers allow detecting areas affected by multi-temporal variations of the radar backscattered signal. Among them, areas potentially related to rapid-moving landslides are identified with a robust statistical analysis. The performance of the implemented procedure was tested in back analysis  in an area of about 3000 km$^2$ in Papua New Guinea. Here, in 2018, two consecutive earthquakes (M7.5 and M6.7) triggered widespread slope failures causing more than 100 fatalities and severe damage to roads and buildings. The simulation of a multi-temporal survey of about seven months, before and after earthquakes, revealed the ability of the implemented procedure to detect statistical evidences of significant land cover changes in correspondence of the two events. Moreover, results demonstrated that the zones  characterized by significant backscattering changes resulted in a reasonable agreement with those affected by landslides , as compared to the ground truth data.

The study highlighted advantages of free SAR products that may guide the scientific community and the local authorities to develop archives of freely accessible data, suitable for implementing streamlines of information aimed to monitor natural and urbanized areas. As demonstrated in the case study, the proposed procedure has the potential to be a valid support in landslide emergency management, providing in near real-time relevant information for civil protection authorities and scientists involved in the emergency response. Future improvements may limit the user decisions in the model parameterization, optimizing the processing times and refining the filtering of landslide-related changes by considering also geological and geomorphological factors.

470 ***Author contributions***. GE and IM implemented the proposed procedure. AM designed the pre-processing of SAR images. GE and IM designed the segmentation procedure. GE and IM carried out the experiments. GE wrote a first draft of the manuscript. MR and GE carried out statistics about landslide-like segments. IM, GE and PR analyzed the results. GE, SS, IM and PR improved the final manuscript version.  475

[revised manuscript text omitted]

**and have the only purpose to differentiate the several segments; d) the extracted landslide-related segments (in blue). Optical images have been downloaded from the Planet explorer application (Planet, 2017).**

650

[Figure]

**Figure 5 - Statistics of the segments identified for each change detection. a) number of segments with more than 5 pixels; b-c) maximum and average number of pixels per segment. For the change detections 9 and 10, the two peaks indicate the occurrence of widespread land cover changes. The dashed lines show the 95° percentiles of the distributions (not including change detections 9 and 10).**

[Figure]

**Figure 6 - The map shows location of the epicenters of the two main earthquakes, and the distribution of segments representing SAR amplitude changes for the change detections 9 and 10. Yellow polygons are areas  affected by clusters of landslides, as interpreted from optical data. The white rectangle identifies the AoI (see Figure 2).**

[Figure]

**Figure 7 - Comparison of segment's areas statistics related to different change detections.**

[Figure]

665     **Figure 8 - Histogram representing the differences,**  **between all the compared segment's areas distributions, calculated according to the p-value of the Kolmogorov-Smirnov tests.**

[Figure]

**Figure 9 - Frequency - area distribution of segments resulted in the change detections 9 (on the left) and 10 (on the right), and fitting with a Double Pareto simplified model.**

---

## Author Response (AR1)

**A semi-automatic spaceborne SAR-based procedure to support the detection of rapid-moving landslides using spaceborne SAR imagery**

Giuseppe Esposito1, Ivan Marchesini2, Alessandro Cesare Mondini2, Paola Reichenbach2, Mauro Rossi2, Simone Sterlacchini3

5 1National Research Council, Research Institute for Geo-Hydrological Protection (CNR-IRPI), Rende (CS), 87036, Italy 2National Research Council, Research Institute for Geo-Hydrological Protection (CNR-IRPI), Perugia, 06128, Italy 3National Research Council, Research Institute for-of Environmental Geology and Geoengineering (CNR-IGAG), Milano, 20126, Italy

Correspondence to: Giuseppe Esposito (giuseppe.esposito@irpi.cnr.it)

[revised manuscript text omitted]

- 85 polarimetric modes) for a target in the form of the scattering matrix (Skriver, 2012). According to Plank et al. (2016), these data provide more information on the ground, which enables a better land cover classification and landslide mapping. Successful applications were described by Yamaguchi (2012), Shimada et al. (2014), Li et al. (2014) and Plank et al. (2016). The use of SAR data to analyze landslides and/or potentially unstable slopes should hence increase, also in relation to a series of valuable technical innovations. The improved revisiting times and spatial resolution of the images, for example,
- 90 represent a key factor during disaster response operations, when a preliminary localization of areas potentially affected by major landslides is crucial. Revisiting times have been in fact reduced from 35 days of ERS and Envisat satellites, to 12 hours (at 40° latitude, in case of emergency response) of the COSMO-SkyMed constellation (Casagli et al., 2017). The enhanced spatial resolution (azimuth or along-track resolution x range or across-track resolution) of images spans in the order of few meters (i.e. 1-10 m), resulting more detailed with respect to the coarser resolution of the first-generation 95 satellites characterized by pixel sizes of 10-30 up to 100 meters (Plank, 2014).
- Among the most advanced SAR spaceborne systems (Casagli et al., 2017), there are those of the mission Sentinel-1 operated by the European Space Agency (ESA) in the frame of the European Union's Copernicus Programme. Satellites Sentinel-1A and 1B acquire images characterized by a spatial resolution pixels with sizes ranging from 5 (range)  $\times$  20 (azimuth) m, in the default acquisition mode for land observations (Interferometric Wide Swath mode - IW), up to 5x5 m, depending on the in

[revised manuscript text omitted]
).5 Such distribution in fact indicatesing a random nature of the LR values, distribution that is typical of no significant when land cover changes are not relevant. We highlight that, in such a case, given that LR values are typically commonly small and positive or negative, the  $\mu$  value is equal or very
- 235 close to zero.

In this way, all the segments of the "average layer" characterized by  $\mu_s$  values larger than  $|\mu^+(2\bar{\sigma})|$  are then extracted and classified. Segments with  $\mu_s$  values lower than a confidence interval of 95% ( $\mu_s

The python-based script (Fig. 1, Data ingestion) is devoted to the automatic querying and downloading of Sentinel-1 SAR images from the ESA Sentinel Data Hub. The script, based on the SentinelSat toolbox (Kersten et al., 2018), is set to query the Sentinel Data Hub with a daily frequency, even though new images may be available every 6 or 12 days, depending on the geographic area.

260

The consecutive group of scripts, written in GNU/Bash programming language (Fig. 1), is aimed at: (i) pre-processing the Sentinel-1 images (section 2.1), (ii) detection of the changes in SAR amplitude and production of Log-Ratio maps (section 2.2), (iii) segmentation of the LR maps (section 2.3) and, (iv) identification of areas potentially affected by land cover changes (section 2.4). This group of scripts is executed automatically when new Sentinel-1 images are available and downloaded by the python-based script.

The bash-scripts require the following settings defined by the user: 1) the path of the folder where the downloaded SAR images are stored; 2) values of the parameters required to use for the segmentation (see section 2.3), and (3) the spatial coordinates of the area of interest (if it is a portion of the downloaded SAR images). No further information is needed since the commands are executed in a unique automatic sequence. To survey the same area for an unlimited time period, all these 270 settings have to be defined only one time for the chain initialization.

275

265

**3 The Papua New Guinea test site**

We selected as test site, an area located in central Papua New Guinea (Fig. 2) that was affected by a severe seismic sequence at the beginning of 2018. On 25 February, the area was hit by a main seismic event (M7.5) followed by several aftershocks, including a M6.7 earthquake on 6 March. The strong mainshock, rather superficial with a hypocentral depth at 23.4 km (USGS, 2018), caused building collapses, road damage and widespread landslides mostly along the Tagari river valley and the slopes of Mount Sisa (McCue et al., 2018). According to the International Federation of Red Cross and Red Crescent Societies (IFRC, 2018), more than 100 people died, most of them due to landslides.

To test the implemented procedure, we have analyzed an area of about 3000 km2 in the mountainous region close to the epicenters of the mainshock (AoI in Fig. 2), where preliminary information on landslides were available (Petley, 2018a, b).

- 280 To simulate a periodic survey covering pre- and post-earthquake periods, we downloaded 36 Sentinel-1 images from the Sentinel Data Hub (https://scihub.copernicus.eu/) acquired along the satellite track n.82 in ascending orbit with a temporal frequency of 12 days, from 12 November 2017 to 6 June 2018. Considering that the majority of the slopes in the study area are exposed towards West, to limit geometrical distortions in the single images and in the change detection estimation, we preferred to use IW-SLC products acquired in ascending mode, with a VV-VH polarization. Each IW product is collected with a swath characterized by a width of 250 km, subdivided in turn to three sub-swaths containing one image per
- polarization, consisting of a series of bursts which are processed as independent SLC images. The downloaded images were used to perform a total of 17 change detection analyses which resulted in likewise LR layers,
  - with a pixel size of about 14 m. The values of the segmentation parameters were defined with an interactive manual analysis (see section 2.3) by segmenting the "pre-post M7.5 earthquake" LR layer, selecting the spatial kernel size (*hs*) of 10 pixels
- (see section 2.3), and setting the maximum number of iterations to 200. This size of the spatial kernel was set to 10 pixels to detect significant differences of LR values during the smoothing stage of the segmentation process, taking into account the approximate expected size of the land cover changes. In the interactive (manual) analysis, we selected bandwidth sizes (*hr*, see section 2.3) ranging from 0.0005 to 0.016, and thresholds (*th*) from 0.001 to 0.016 (Fig. 3), obtaining 20 different parameter combinations. For each couple of parameters, the number of generated segments and their average size were
- 295 plotted as shown-in Fig-ure 3. Points highlight the major impact of the *hr* parameter with respect to the role played by the threshold (*th*) parameter, in defining the number of total generated segments. Below an *hr* value of 0.004 over segmentation occur, whereas for *hr* values equal or larger than 0.004, the number of generated segments tends to become small and constant. With the aim of avoiding over segmentation while maintaining a reasonable average size of the segments (to be able to delineate also small patches of the terrain where changes occurred), and considering a visual inspection of the
- 300 segmentation results obtained with the different combinations, we decided to run i.segment in the semi-automatic processing chain using the following set of parameters values: *hs*=10, *hr*=0.004, *th*=0.008, *minsize*=2, *iterations*=200 (see section 2.3).
   After the segmentation of the 17 LR layers, areas affected by layover and shadowing effects were masked out in order to avoid errors in the statistical analysis described below and in the localization of potential landslides. The mask was developed in SNAP by means of the SAR Simulation Terrain Correction tool, exploiting the SRTM 1Sec DEM.
- 305 #The segments with a minimum size of 5 pixels were extracted in the area of interest (an example is shown in Fig. 4d), and statistics were calculated according to the confidence intervals described in the methodology section. We decided to select only the segments with a minimum size of 5 pixels, corresponding to a minimum area of about 980 m2 (i.e. a single-pixel area roughly equal to 196 m2 times 5 pixels), after a rough-general evaluation of the preliminary landslide-related images published on news websites and social networks, and considering that the occurrence detection of smaller mass

310 movements segments in the test area were was not significant at the scale of our analysis for detecting landslides. It is worth noting that accuracy of such a minimum area is not accurate due to the use of a geographic (not projected) reference system (WGS84).

In Fig-ure 5, statistics of the selected segments are displayed for each change detection. The analysis of the histograms revealed that two main peaks occurred for corresponding to the change detections 9 and 10. Change detection 9 considers

- 315 images acquired before and after the M7.5 earthquake, whereas change detection 10 the images acquired on 28 February and 12 March 2018. The first peak highlights widespread changes related also to landslides extensively documented after the M7.5 event (Petley, 2018a). The second peak was instead unexpected and was probably due to the occurrence of further landslides triggered by the M6.7 event on 6 March 2018. In Fig-ure 6, segments related to these two peaks are displayed (red pixels = change detection 9; blue pixels = change detection 10). To check whether these segments were effectively located in
- 320 areas characterized by a high concentration of seismic-induced landslides, we analyzed optical images available on the Planet explorer application (Planet, 2017). By means of a visual interpretation, we identified the zones (the yellow polygons shown in Fig. 6) where clusters of landslides occurred, verifying a general accordance with the spatial distribution of both red and blue segments. The map shows that the two groups of segments are in general accordance with the areas (in yellow) really affected by landslides, as interpreted from the optical images available on the Planet explorer application (Planet, 2017).

325 <del>2017).</del>

[revised manuscript text omitted]

**365 **4** Discussion**

In this article, we describe a semi-automatic processing chain aimed at identifying SAR amplitude changes that can be partially explained by the occurrence of rapid-mass movements. We have selected SAR data since they have the advantage to be not affected by the cloud cover disturbance. In fact, as described by Mondini et al. (2019), the use of SAR amplitude data can mitigate the cloud coverage issue and can allow detecting landslides that, otherwise, might remain unknown or

370

unnoticed for a long time. In this way, the procedure can be exploited for a "continuous", in terms of time, slope monitoring activity, even if failures occur during long-lasting periods of precipitation and persistent cloud cover that do not allow to use optical data for a rapid and detailed landslide recognition. In the selected study area, a widespread cloud cover persisted for several weeks during and after the seismic sequence. The first cloudless optical image of the area damaged by the seismic shaking was published by the daily monitoring service delivered by ©2019 Planet Labs Inc. (www.planet.com) on 25 March,

- 375 almost one month after the M7.5 mainshock that triggered numerous landslides. The high cloud persistence is quite common in Papua New Guinea, and in fact this is included in the cloudiest regions of the world with annual cloud frequency (proportion of days with a positive cloud flag) higher than eighty percent (Wilson and Jets, 2016; Mondini et al., 2019). As consequence, the use of optical data in this area, and in other mountainous regions exposed to prolonged rainfall related to monsoons, cyclones or other persistent meteorological systems results tricky.
- 380 The obtained results depend on the definition of the image pre-processing and segmentation parameters that should be calibrated a priori (see sections 2.3 and 2.4). While images geometric and radiometric corrections are quite standard and well-accepted procedures, the SAR multiplicative noise filtering remains a largely discussed point in the scientific literature and there is not a consensus on the selection of strategies. We choose the Frost filter because it already proved to be properly working in mountainous environments (Schellenberg et al., 2012) and it was used successfully in previous studies dealing
- 385 with landslides (Mondini, 2017). We acknowledge that different filters might have brought different results or requested a different tuning of the segmentation procedure. The impact of different filters on our procedure might be an interesting follow up of this work. Another improvement may consist in the use of images acquired in both ascending and descending geometries. The use of ascending images was only related to focus this first step of the work on the implementation of the entire processing chain, that we tried to simplify as much as possible. In fact, there is no doubt that combining images
- 390 acquired in ascending and descending geometries can improve the quality of results, representing a non-trivial advancement of the procedure that was out of the aim of this first implementation. The a priori choice of using ascending products was based on the findings that most of the slopes in the study area are exposed towards West, with the aim of limiting the inclusion of geometrical distortions in the change detection products.
- The tuning of the segmentation parameters is the key element for identifying areas affected by significant land cover 395 changes, also induced by rapid-moving slope movements. This process can be retained event-dependent, requiring a wellknown landslide event occurred in the past in the analyzed area or in zones with similar topographic and land use characteristics. In the case study here described, definite values of the segmentation parameters were obtained by segmenting the pre-post M7.5 earthquake LR layer, and by testing different values combinations (Fig. 3). This may represent a limit of the proposed procedure if one would apply it in different geomorphic settings without past landslide events, or identifying 400 different types of slope failures. On the other hand, if a proper event-based tuning operation is performed, a continuous monitoring of slopes can be efficiently earry-carried out without temporal limitations, exploiting both pre- and post-event available images, as done in the current case history. The described application highlighted in fact that by keeping the same parameters values, landslides and other land cover changes triggered by the M6.7 aftershock were also detected. Overlapping between the calculated segments (i.e. change detections 9 and 10) to ground truth data revealed that largest SAR amplitude
- 405 changes corresponded often with landslides (Fig. 6). A further evidence was provided by the statistical distributions shown in Figure 9, resulted similar to those estimated by other landslide-related studies (Stark and Hovius, 2001; Malamud et al., 2004; Rossi et al., 2012; Schlögel et al., 2015). The segments located mostly outside the areas affected by landslides were caused instead by other land cover changes that were 
[revised manuscript text omitted]
. GE and IM implemented the proposed procedure. AM designed the pre-processing of SAR images. GE and IM designed the segmentation procedure. GE and IM carried out the experiments. GE wrote a first draft of the manuscript. MR and GE carried out statistics about landslide-like segments. IM, GE and PR analyzed the results. GE, SS, IM and PR improved the final manuscript version. IM, GE and AM designed and realized the processing chain. GE and IM earried out the experiments. GE wrote a first draft of the manuscript. MR and GE carried out statistics about landslide-like segments. IM, GE and PR analyzed the results. GE, SS, IM and PR improved the final manuscript version. GE, SS, IM and PR improved the final manuscript version.

[revised manuscript text omitted]